# Learning with Bandit Feedback in Potential Games

**Johanne Cohen**
LRI-CNRS, Université Paris-Sud,Université Paris-Saclay, France
`johanne.cohen@lri.fr`

**Amélie Héliou**
LIX, Ecole Polytechnique, CNRS, AMIBio, Inria, Université Paris-Saclay
`amelie.heliou@polytechnique.edu`

**Panayotis Mertikopoulos**
Univ. Grenoble Alpes, CNRS, Inria, LIG, F-38000, Grenoble, France
`panayotis.mertikopoulos@imag.fr`

## Abstract

This paper examines the equilibrium convergence properties of no-regret learning with exponential weights in potential games. To establish convergence with minimal information requirements on the players' side, we focus on two frameworks: the *semi-bandit case* (where players have access to a noisy estimate of their payoff vectors, including strategies they did not play), and the *bandit case* (where players are only able to observe their in-game, realized payoffs). In the semi-bandit case, we show that the induced sequence of play converges almost surely to a Nash equilibrium at a quasi-exponential rate. In the bandit case, the same result holds for $\varepsilon$-approximations of Nash equilibria if we introduce an exploration factor $\varepsilon > 0$ that guarantees that action choice probabilities never fall below $\varepsilon$. In particular, if the algorithm is run with a suitably decreasing exploration factor, the sequence of play converges to a bona fide Nash equilibrium with probability 1.

## 1   Introduction

Given the manifest complexity of computing Nash equilibria, a central question that arises is whether such outcomes could result from a dynamic process in which players act on empirical information on their strategies' performance over time. This question becomes particularly important when the players' view of the game is obstructed by situational uncertainty and the "fog of war": for instance, when deciding which route to take to work each morning, a commuter is typically unaware of how many other commuters there are at any given moment, what their possible strategies are, how to best respond to their choices, etc. In fact, in situations of this kind, players may not even know that they are involved in a game; as such, it does not seem reasonable to assume full rationality, common knowledge of rationality, flawless execution, etc. to justify the Nash equilibrium prediction.

A compelling alternative to this "rationalistic" viewpoint is provided by the framework of *online learning*, where players are treated as oblivious entities facing a repeated decision process with a priori unknown rules and outcomes. In this context, when the players have no Bayesian prior on their environment, the most widely used performance criterion is that of *regret minimization*, a worst-case guarantee that was first introduced by Hannan [1], and which has given rise to a vigorous literature at the interface of optimization, statistics and theoretical computer science – for a survey, see [2, 3]. By this token, our starting point in this paper is the following question:

> *If all players of a repeated game follow a no-regret algorithm,*
> *does the induced sequence of play converge to Nash equilibrium?*

For concreteness, we focus on the *exponential weights* (EW) scheme [4–7], one of the most popular and widely studied algorithms for no-regret learning. In a nutshell, the main idea of the method is that the optimizing agent tallies the cumulative payoffs of each action and then employs a pure strategy with probability proportional to the exponential of these cumulative "scores". Under this scheme, players are guaranteed a universal, min-max $\mathcal{O}(T^{1/2})$ regret bound (with $T$ denoting the horizon of play), and their empirical frequency of play is known to converge to the game's set of *coarse correlated equilibria* (CCE) [8].

In this way, no-regret learning would seem to provide a positive partial answer to our original question: coarse correlated equilibria are indeed learnable if all players follow an exponential weights learning scheme. On the flip side however, the set of coarse correlated equilibria may contain highly non-rationalizable strategies, so the end prediction of empirical convergence to such equilibria is fairly lax. For instance, in a recent paper, Viossat and Zapechelnyuk constructed a $4 \times 4$ variant of Rock-Paper-Scissors with a coarse correlated equilibrium that assigns positive weight *only* on strictly dominated strategies [9]. Even more recently, [10] showed that the mean dynamics of the exponential weights method (and, more generally, any method "following the regularized leader") may cycle in perpetuity in zero-sum games, precluding any possibility of convergence to equilibrium in this case. Thus, in view of these negative results, a more calibrated answer to the above question is "not always": especially when the issue at hand is convergence to a Nash equilibrium (as opposed to coarser notions), "no regret" is a rather loose guarantee.

**Paper outline and summary of results.**    To address the above limitations, we focus on two issues:

*a*)  Convergence to Nash equilibrium (as opposed to correlated equilibria, coarse or otherwise).

*b*)  The convergence of the actual sequence of play (as opposed to empirical frequencies).

The reason for focusing on the actual sequence of play is that time-averages provide a fairly weak convergence mode: a priori, a player could oscillate between non-equilibrium strategies with suboptimal payoffs, but time-averages might still converge to equilibrium. On the other hand, convergence of the actual sequence of play both implies empirical convergence and also guarantees that players will be playing a Nash equilibrium in the long run, so it is a much stronger notion.

To establish convergence, we focus throughout on the class of *potential games* [11] that has found widespread applications in theoretical computer science [12], transportation networks [13], wireless communications [14], biology [15], and many other fields. We then focus on two different feedback models: in the *semi-bandit framework* (Section 3), players are assumed to have some (possibly imperfect) estimate of their payoff vectors at each stage, including strategies that they did not play; in the full *bandit framework* (Section 4), this assumption is relaxed and players are only assumed to observe their realized, in-game payoff at each stage.

Starting with the semi-bandit case, our main result is that under fairly mild conditions for the errors affecting the players' observations (zero-mean martingale noise with tame second-moment tails), learning with exponential weights converges to a Nash equilibrium of the game with probability $1$ (or to an $\varepsilon$-equilibrium if the algorithm is implemented with a uniform exploration factor $\varepsilon > 0$).[1] We also show that this convergence occurs at a *quasi-exponential rate*, i.e. much faster than the algorithm's $\mathcal{O}(\sqrt{T})$ regret minimization rate would suggest.

These conclusions also apply to the bandit framework when the algorithm is run with a positive exploration factor $\varepsilon > 0$. Thus, by choosing a sufficiently small exploration factor, the end state of the EW algorithm in potential games with bandit feedback is arbitrarily close to a Nash equilibrium.

On the other hand, extending the stochastic approximation and martingale limit arguments that underlie the bandit analysis to the $\varepsilon = 0$ case is not straightforward. However, by letting the exploration factor go to zero at a suitable rate (similar to the temperature parameter in simulated annealing schemes), we are able to recover convergence to the game's exact Nash set (and not an approximation thereof). We find this property particularly appealing for practical applications because it shows that equilibrium can be achieved in a wide class of games with minimal information requirements.

**Related work.** No-regret learning has given rise to a vast corpus of literature in theoretical computer science and machine learning, and several well-known families of algorithms have been proposed for that purpose. The most popular of these methods is based on exponential/multiplicative weight update rules, and several variants of this general scheme have been studied under different names in the literature (Hedge, EXP3, etc.) [4–7].

When applied to games, the time-average of the resulting trajectory of play converges to equilibrium in two-player zero-sum games [6, 16, 17] and the players' social welfare approaches an approximate optimum [18]. In a similar vein, focusing on the so-called "Hedge" variant of the multiplicative weights (MW) algorithm, Kleinberg et al. [19] proved that the dynamics' long-term limit in load balancing games is exponentially better than the worst correlated equilibrium. The convergence rate to approximate efficiency and to coarse correlated equilibria was further improved by Syrgkanis et al. [20] for a wide class of $N$-player normal form games using a natural class of regularized learning algorithms. This result was then extended to a class of games known as *smooth games* [21] with good properties in terms of the game's price of anarchy [22].

In the context of potential games, learning algorithms and dynamics have received signifcant attention and considerable efforts have been devoted to studying the long-term properties of the players' actual sequence of play. To that end, Kleinberg et al. [23] showed that, after a polynomially small transient stage, players end up playing a pure equilibrium for a fraction of time that is arbitrarily close to $1$ with probability also arbitrarily close to $1$. Mehta et al. [24] obtained a stronger result for (generic) 2-player coordination games, showing that the multiplicative weights algorithm (a linearized variant of the EW algorithm) converges to a pure Nash equilibrium for all but a measure $0$ of initial conditions. More recently, Palaiopanos et al. [25] showed that the MW update rule converges to equilibrium in potential games; however, if the EW algorithm is run with a constant step-size that is not small enough, the induced sequence of play may exhibit chaotic behavior, even in simple $2 \times 2$ games. On the other hand, if the same algorithm is run with a decreasing step-size, Krichene et al. [26] showed that play converges to Nash equilibrium in all nonatomic potential games with a convex potential (and hence, in all nonatomic congestion games).

In the above works, players are assumed to have full (though possibly imperfect) knowledge of their payoff vectors, including actions that were not chosen. Going beyond this semi-bandit framework, Coucheney et al. [27] showed that a "penalty-regulated" variant of the EW algorithm converges to $\varepsilon$-logit equilibria (and hence $\varepsilon$-approximate Nash equilibria) in congestion games with bandit feedback. As in [26], the results of Coucheney et al. [27] employ the powerful ordinary differential equation (ODE) method of Benaïm [28] which leverages the convergence of an underlying, continuous-time dynamical system to obtain convergence of the algorithm at hand. We also employ this method to compare the actual sequence of play to the replicator dynamics of evolutionary game theory [29]; however, finetuning the bias-variance trade-off that arises when estimating the payoff of actions that were not employed is a crucial difficulty in this case. Overcoming this hurdle is necessary when seeking convergence to actual Nash equilibria (as opposed to $\varepsilon$-approximations thereof), so a key contribution of our paper is an extension of Benaïm's theory to account for estimators with (possibly) unbounded variance.

## 2   The setup

### 2.1   Game-theoretic preliminaries

An $N$-player *game in normal form* consists of a (finite) set of *players* $\mathcal{N} = \{1, \dots, N\}$, each with a finite set of *actions* (or *pure strategies*) $\mathcal{A}_i$. The preferences of the $i$-th player for one action over another are determined by an associated *payoff function* $u_i \colon \mathcal{A} \equiv \prod_i \mathcal{A}_i \to \mathbb{R}$ that maps the *profile* $(\alpha_i; \alpha_{-i})$ of all players' actions to the player's reward $u_i(\alpha_i; \alpha_{-i})$.[2] Putting all this together, a game will be denoted by the tuple $\Gamma \equiv \Gamma(\mathcal{N}, \mathcal{A}, u)$.

Players can also *mix* their strategies by playing probability distributions $x_i = (x_{i\alpha_i})_{\alpha_i \in \mathcal{A}_i} \in \Delta(\mathcal{A}_i)$ over their action sets $\mathcal{A}_i$. The resulting probability vector $x_i$ is called a *mixed strategy* and we write $\mathcal{X}_i = \Delta(\mathcal{A}_i)$ for the mixed strategy space of player $i$. Aggregating over players, we also write $\mathcal{X} = \prod_i \mathcal{X}_i$ for the game's *strategy space*, i.e. the space of all mixed strategy profiles $x = (x_i)_{i \in \mathcal{N}}$.

In this context (and in a slight abuse of notation), the expected payoff of the $i$-th player in the profile $x = (x_1, \ldots, x_N)$ is

$$u_i(x) = \sum_{\alpha_1 \in \mathcal{A}_1} \cdots \sum_{\alpha_N \in \mathcal{A}_N} u_i(\alpha_1, \ldots, \alpha_N) \, x_{1\alpha_1} \cdots x_{N\alpha_N}. \tag{2.1}$$

To keep track of the payoff of each pure strategy, we also write $v_{i\alpha_i}(x) = u_i(\alpha_i; x_{-i})$ for the payoff of strategy $\alpha_i \in \mathcal{A}_i$ under the profile $x \in \mathcal{X}$ and

$$v_i(x) = (v_{i\alpha_i}(x))_{\alpha_i \in \mathcal{A}_i} \tag{2.2}$$

for the resulting *payoff vector* of player $i$. We thus have

$$u_i(x) = \langle v_i(x), x_i \rangle = \sum_{\alpha_i \in \mathcal{A}_i} x_{i\alpha_i} v_{i\alpha_i}(x), \tag{2.3}$$

where $\langle v, x \rangle \equiv v^\top x$ denotes the ordinary pairing between $v$ and $x$.

The most widely used solution concept in game theory is that of a *Nash equilibrium* (NE), i.e. a state $x^* \in \mathcal{X}$ such that

$$u_i(x_i^*; x_{-i}^*) \geq u_i(x_i; x_{-i}^*) \quad \text{for every deviation } x_i \in \mathcal{X}_i \text{ of player } i \text{ and all } i \in \mathcal{N}. \tag{NE}$$

Equivalently, writing $\text{supp}(x_i) = \{\alpha_i \in \mathcal{A}_i : x_i > 0\}$ for the support of $x_i \in \mathcal{X}_i$, we have the characterization

$$v_{i\alpha_i}(x^*) \geq v_{i\beta_i}(x^*) \quad \text{for all } \alpha_i \in \text{supp}(x_i^*) \text{ and all } \beta_i \in \mathcal{A}_i, i \in \mathcal{N}. \tag{2.4}$$

A Nash equilibrium $x^* \in \mathcal{X}$ is further said to be *pure* if $\text{supp}(x_i^*) = \{\hat{\alpha}_i\}$ for some $\hat{\alpha}_i \in \mathcal{A}_i$ and all $i \in \mathcal{N}$. In *generic games* (that is, games where small changes to any payoff do not introduce new Nash equilibria or destroy existing ones), every pure Nash equilibrium is also *strict* in the sense that (2.4) holds as a strict inequality for all $\alpha_i \neq \hat{\alpha}_i$.

In our analysis, it will be important to consider the following relaxations of the notion of a Nash equilibrium: First, weakening the inequality (NE) leads to the notion of a $\delta$-*equilibrium*, defined here as any mixed strategy profile $x^* \in \mathcal{X}$ such that

$$u_i(x_i^*; x_{-i}^*) + \delta \geq u_i(x_i; x_{-i}^*) \quad \text{for every deviation } x_i \in \mathcal{X}_i \text{ and all } i \in \mathcal{N}. \tag{NE$_\delta$}$$

Finally, we say that $x^*$ is a *restricted equilibrium* (RE) of $\Gamma$ if

$$v_{i\alpha_i}(x^*) \geq v_{i\beta_i}(x^*) \quad \text{for all } \alpha_i \in \text{supp}(x_i^*) \text{ and all } \beta_i \in \mathcal{A}_i', i \in \mathcal{N}, \tag{RE}$$

where $\mathcal{A}_i'$ is some restricted subset of $\mathcal{A}_i$ containing $\text{supp}(x_i^*)$. In words, restricted equilibria are Nash equilibria of $\Gamma$ restricted to subgames where only a subset of the players' pure strategies are available at any given moment. Clearly, Nash equilibria are restricted equilibria but the converse does not hold: for instance, every pure strategy profile is a restricted equilibrium, but not necessarily a Nash equilibrium.

Throughout this paper, we will focus almost exclusively on the class of *potential games*, which have been studied extensively in the context of congestion, traffic networks, oligopolies, etc. Following Monderer and Shapley [11], $\Gamma$ is a *potential game* if it admits a *potential function* $f : \prod_i \mathcal{A}_i \to \mathbb{R}$ such that

$$u_i(x_i; x_{-i}) - u_i(x_i'; x_{-i}) = f(x_i; x_{-i}) - f(x_i'; x_{-i}), \tag{2.5}$$

for all $x_i, x_i' \in \mathcal{X}_i$, $x_{-i} \in \mathcal{X}_{-i} \equiv \prod_{j \neq i} \mathcal{X}_i$, and all $i \in \mathcal{N}$. A simple differentiation of (2.1) then yields

$$v_i(x) = \nabla_{x_i} u_i(x) = \nabla_{x_i} f(x) \quad \text{for all } i \in \mathcal{N}. \tag{2.6}$$

Obviously, every local maximizer of $f$ is a Nash equilibrium so potential games always admit Nash equilibria in pure strategies (which are also strict if the game is generic).

## 2.2 The exponential weights algorithm

Our basic learning framework is as follows: At each stage $n = 1, 2, \ldots$, all players $i \in \mathcal{N}$ select an action $\alpha_i(n) \in \mathcal{A}_i$ based on their mixed strategies; subsequently, they receive some feedback on their chosen actions, they update their mixed strategies, and the process repeats.

A popular (and very widely studied) class of algorithms for no-regret learning in this setting is the *exponential weights* (EW) scheme introduced by Vovk [4] and studied further by Auer et al. [5], Freund and Schapire [6], Arora et al. [7], and many others. Somewhat informally, the main idea is that each player tallies the cumulative payoffs of each of their actions, and then employs a pure strategy $\alpha_i \in \mathcal{A}_i$ with probability roughly proportional to the these cumulative payoff "scores". Focusing on the so-called "$\varepsilon$-HEDGE" variant of the EW algorithm [6], this process can be described in pseudocode form as follows:

---

**Algorithm 1** $\varepsilon$-HEDGE with generic feedback

---

**Require:** step-size sequence $\gamma_n > 0$, exploration factor $\varepsilon \in [0, 1]$, initial scores $Y_i \in \mathbb{R}^{\mathcal{A}_i}$.

1: **for** $n = 1, 2, \ldots$ **do**
2:    **for** every player $i \in \mathcal{N}$ **do**
3:       set mixed strategy: $X_i \leftarrow \varepsilon \, \mathrm{unif}_i + (1 - \varepsilon) \, \Lambda_i(Y_i)$;
4:       choose action $\alpha_i \sim X_i$;
5:       acquire estimate $\hat{v}_i$ of realized payoff vector $v_i(\alpha_i; \alpha_{-i})$;
6:       update scores: $Y_i \leftarrow Y_i + \gamma_n \hat{v}_i$;
7:    **end for**
8: **end for**

---

Mathematically, Algorithm 1 represents the recursion

$$X_i(n) = \varepsilon \, \mathrm{unif}_i + (1 - \varepsilon) \, \Lambda_i(Y_i(n)),$$
$$Y_i(n+1) = Y_i(n) + \gamma_{n+1} \hat{v}_i(n+1), \tag{$\varepsilon$-Hedge}$$

where

$$\mathrm{unif}_i = \frac{1}{|\mathcal{A}_i|}(1, \ldots, 1) \tag{2.7}$$

stands for the uniform distribution over $\mathcal{A}_i$ and $\Lambda_i \colon \mathbb{R}^{\mathcal{A}_i} \to \mathcal{X}_i$ denotes the *logit choice map*

$$\Lambda_i(y_i) = \frac{(\exp(y_{i\alpha_i}))_{\alpha_i \in \mathcal{A}_i}}{\sum_{\alpha_i \in \mathcal{A}_i} \exp(y_{i\alpha_i})}, \tag{2.8}$$

which assigns exponentially higher probability to pure strategies with higher scores. Thus, action selection probabilities under ($\varepsilon$-Hedge) are a convex combination of uniform exploration (with total weight $\varepsilon$) and exponential weights (with total weight $1 - \varepsilon$).[3] As a result, for $\varepsilon \approx 1$, action selection is essentially uniform; at the other extreme, when $\varepsilon = 0$, we obtain the original Hedge algorithm of Freund and Schapire [6] with feedback sequence $\hat{v}(n)$ and no explicit exploration.

The no-regret properties of ($\varepsilon$-Hedge) have been extensively studied in the literature as a function of the algorithm's step-size sequence $\gamma_n$, exploration factor $\varepsilon$, and the statistical properties of the payoff estimates $\hat{v}(n)$ – for a survey, we refer the reader to [2, 3]. In our convergence analysis, we examine the role of each of these factors in detail, focusing in particular on the distinction between "*semi-bandit feedback*" (when it is possible to estimate the payoff of pure strategies that were not played) and "*bandit feedback*" (when players only observe the payoff of their chosen action).

## 3 Learning with semi-bandit feedback

### 3.1 The model

We begin with the *semi-bandit framework*, i.e. the case where each player has access to a possibly imperfect estimate of their entire payoff vector at stage $n$. More precisely, we assume here that the feedback sequence $\hat{v}_i(n)$ to Algorithm 1 is of the general form

$$\hat{v}_i(n) = v_i(\alpha_i(n); \alpha_{-i}(n)) + \xi_i(n), \tag{3.1}$$

where $(\xi_i(n))_{i \in \mathcal{N}}$ is a martingale noise process representing the players' estimation error and satisfying the following statistical hypotheses:

1. *Zero-mean:*

$$\mathbb{E}[\xi_i(n) \,|\, \mathcal{F}_{n-1}] = 0 \quad \text{for all } n = 1, 2, \dots \text{ (a.s.).} \qquad \text{(H1)}$$

2. *Tame tails:*

$$\mathbb{P}(\|\xi_i(n)\|_\infty^2 \geq z \,|\, \mathcal{F}_{n-1}) \leq A/z^q \quad \text{for some } q > 2, A > 0, \text{ and all } n = 1, 2, \dots \text{ (a.s.).} \qquad \text{(H2)}$$

In the above, the expectation $\mathbb{E}[\,\cdot\,]$ is taken with respect to some underlying filtered probability space $(\Omega, \mathcal{F}, (\mathcal{F}_n)_{n\in\mathbb{N}}, \mathbb{P})$ which serves as a stochastic basis for the process $(\alpha(n), \hat{v}(n), Y(n), X(n))_{n\geq 1}$.[4] In words, Hypothesis (H1) simply means that the players' feedback sequence $\hat{v}(n)$ is *conditionally unbiased* with respect to the history of play, i.e.

$$\mathbb{E}[\hat{v}_i(n) \,|\, \mathcal{F}_{n-1}] = v_i(X(n-1)), \quad \text{for all } n = 1, 2, \dots \text{ (a.s.).} \qquad (3.2a)$$

Hypothesis (H2) further implies that the variance of the estimator $\hat{v}$ is conditionally bounded, i.e.

$$\mathrm{Var}[\hat{v}(n) \,|\, \mathcal{F}_{n-1}] \leq \sigma^2 \quad \text{for all } n = 1, 2, \dots \text{ (a.s.).} \qquad (3.2b)$$

By Chebyshev's inequality, an estimator with finite variance enjoys the tail bound $\mathbb{P}(\|\xi_i(n)\|_\infty \geq z \,|\, \mathcal{F}_{n-1}) = \mathcal{O}(1/z^2)$. At the expense of working with slightly more conservative step-size policies (see below), much of our analysis goes through with this weaker requirement for the tails of $\xi$. However, the extra control provided by the $\mathcal{O}(1/z^q)$ tail bound simplifies the presentation considerably, so we do not consider this relaxation here. In any event, Hypothesis (H2) is satisfied by a broad range of error noise distributions (including all compactly supported, sub-Gaussian and sub-exponential distributions), so the loss in generality is small compared to the gain in clarity and concision.

## 3.2 Convergence analysis

With all this at hand, our main result for the convergence of ($\varepsilon$-Hedge) with semi-bandit feedback of the form (3.1) is as follows:

**Theorem 1.** *Let $\Gamma$ be a generic potential game and suppose that Algorithm 1 is run with i) semi-bandit feedback satisfying (H1) and (H2); ii) a nonnegative exploration factor $\varepsilon \geq 0$; and iii) a step-size sequence of the form $\gamma_n \propto 1/n^\beta$ for some $\beta \in (1/q, 1]$. Then:*

1. *$X(n)$ converges (a.s.) to a $\delta$-equilibrium of $\Gamma$ with $\delta \equiv \delta(\varepsilon) \to 0$ as $\varepsilon \to 0$.*

2. *If $\lim_{n\to\infty} X(n)$ is an $\varepsilon$-pure state of the form $x_i^* = \varepsilon \, \mathrm{unif}_i + (1-\varepsilon)e_{\hat{\alpha}_i}$ for some $\hat{\alpha} \in \mathcal{A}$, then $\hat{\alpha}$ is a.s. a strict equilibrium of $\Gamma$ and convergence occurs at a quasi-exponential rate:*

$$X_{i\hat{\alpha}_i}(n) \geq 1 - \varepsilon - be^{-c\sum_{k=1}^n \gamma_k} \quad \text{for some positive } b, c > 0. \qquad (3.3)$$

**Corollary 2.** *If Algorithm 1 is run with assumptions as above and no exploration ($\varepsilon = 0$), $X(n)$ converges to a Nash equilibrium with probability 1. Moreover, if the limit of $X(n)$ is pure and $\beta < 1$, we have*

$$X_{i\hat{\alpha}_i}(n) \geq 1 - be^{-cn^{1-\beta}} \quad \text{for some positive } b, c > 0. \qquad (3.4)$$

*Sketch of the proof.* The proof of Theorem 1 is fairly convoluted, so we relegate the details to the paper's technical appendix and only present here a short sketch thereof.

Our main tool is the so-called *ordinary differential equation* (ODE) method, a powerful stochastic approximation scheme due to Benaïm and Hirsch [28, 30]. The key observation is that the mixed strategy sequence $X(n)$ generated by Algorithm 1 can be viewed as a "Robbins–Monro approximation" (an *asymptotic pseudotrajectory* to be precise) of the $\varepsilon$-perturbed exponential learning dynamics

$$\begin{aligned} \dot{y}_i &= v_i(x), \\ x_i &= \varepsilon \, \mathrm{unif}_i + (1-\varepsilon)\, \Lambda_i(y_i), \end{aligned} \qquad (\text{XL}_\varepsilon)$$

By differentiating, it follows that $x_i(t)$ evolves according to the $\varepsilon$-*perturbed replicator dynamics*

$$\dot{x}_{i\alpha} = \left(x_{i\alpha} - |\mathcal{A}_i|^{-1}\varepsilon\right)\left[v_{i\alpha}(x) - (1-\varepsilon)^{-1}\sum_{\beta\in\mathcal{A}_i}(x_{i\beta} - |\mathcal{A}_i|^{-1}\varepsilon)v_{i\beta}(x)\right], \qquad (\text{RD}_\varepsilon)$$

which, for $\varepsilon = 0$, boil down to the ordinary *replicator dynamics* of Taylor and Jonker [29]:

$$\dot{x}_{i\alpha} = x_{i\alpha}[v_{i\alpha}(x) - \langle v_i(x), x_i \rangle], \qquad \text{(RD)}$$

A key property of the replicator dynamics that readily extends to the $\varepsilon$-perturbed variant (RD$_\varepsilon$) is that the game's potential $f$ is a strict Lyapunov function – i.e. $f(x(t))$ is increasing under (RD$_\varepsilon$) unless $x(t)$ is stationary. By a standard result of Benaïm [28], this implies that the discrete-time process $X(n)$ converges (a.s.) to a connected set of rest points of (RD$_\varepsilon$), which are themselves approximate restricted equilibria of $\Gamma$.

Of course, since every $\varepsilon$-pure point of the form $(\varepsilon \, \text{unif}_i + (1 - \varepsilon)e_{\alpha_i})_{i \in \mathcal{N}}$ is also stationary under (RD$_\varepsilon$), the above does not imply that the limit of $X(n)$ is an approximate *equilibrium* of $\Gamma$. To rule out non-equilibrium outcomes, we first note that the set of rest points of (RD$_\varepsilon$) is finite (by genericity), so $X(n)$ must converge to a point. Then, the final step of our convergence proof is provided by a martingale recurrence argument which shows that when $X(n)$ converges to a point, this limit must be an approximate equilibrium of $\Gamma$. Finally, the rate of convergence (3.3) is obtained by comparing the payoff of a player's equilibrium strategy to that of the player's other strategies, and then "inverting" the logit choice map to translate this into an exponential decay rate for $\|X_{i\hat{\alpha}_i}(n) - x^*\|$. □

We close this section with two remarks on Theorem 1. First, we note that there is an inverse relationship between the tail exponent $q$ in (H2) and the decay rate $\beta$ of the algorithm's step-size sequence $\gamma_n \propto n^{-\beta}$. Specifically, higher values of $q$ imply that the noise in the players' observations is smaller (on average and with high probability), so players can be more aggressive in their choice of step-size. This is reflected in the lower bound $1/q$ for $\beta$ and the fact that the players' rate of convergence to Nash equilibrium increases for smaller $\beta$; in particular, (3.3) shows that Algorithm 1 enjoys a convergence bound which is just shy of $\mathcal{O}(\exp(-n^{1-1/q}))$. Thus, if the noise process $\xi$ is sub-Gaussian/sub-exponential (so $q$ can be taken arbitrarily large), a near-constant step-size sequence (small $\beta$) yields an almost linear convergence rate.

Second, if the noise process $\xi$ is "isotropic" in the sense of Benaïm [28, Thm. 9.1], the instability of non-pure Nash equilibria under the replicator dynamics can be used to show that the limit of $X(n)$ is pure with probability 1.[5] When this is the case, the quasi-exponential convergence rate (3.3) becomes universal in that it holds with probability 1 (as opposed to conditioning on $\lim_{n \to \infty} X(n)$ being pure). We find this property particularly appealing for practical applications because it shows that equilibrium is reached *exponentially faster* than the $\mathcal{O}(1/\sqrt{n})$ worst-case regret bound of ($\varepsilon$-Hedge) would suggest.

## 4 Payoff-based learning: the bandit case

We now turn to the *bandit framework*, a minimal-information setting where, at each stage of the process, players only observe their realized payoffs

$$\hat{u}_i(n) = u_i(\alpha_i(n); \alpha_{-i}(n)). \qquad (4.1)$$

In this case, players have no clue about the payoffs of strategies that were not chosen, so they must *construct* an estimator for their payoff vector, including its missing components. A standard way to do this is via the *bandit estimator*

$$\hat{v}_{i\alpha_i}(n) = \frac{\mathbb{1}(\alpha_i(n) = \alpha_i)}{\mathbb{P}(\alpha_i(n) = \alpha_i \mid \mathcal{F}_{n-1})} \cdot \hat{u}_i(n) = \begin{cases} \hat{u}_i(n)/X_{i\alpha_i}(n-1) & \text{if } \alpha_i = \alpha_i(n), \\ 0 & \text{otherwise.} \end{cases} \qquad (4.2)$$

Indeed, a straightforward calculation shows that

$$\begin{aligned} \mathbb{E}[\hat{v}_{i\alpha_i}(n) \mid \mathcal{F}_{n-1}] &= \sum_{\alpha_{-i} \in \mathcal{A}_{-i}} X_{-i,\alpha_{-i}}(n-1) \sum_{\beta_i \in \mathcal{A}_i} X_{i\beta_i}(n-1) \frac{\mathbb{1}(\alpha_i = \beta_i)}{X_{i\alpha_i}(n-1)} u_i(\beta_i; \alpha_{-i}) \\ &= u_i(\alpha_i; X_{-i}(n-1)) \\ &= v_{i\alpha_i}(X(n-1)), \end{aligned} \qquad (4.3)$$

so the estimator (4.2) is unbiased in the sense of (H1)/(3.2a). On the other hand, a similar calculation shows that the variance of $\hat{v}_{i\alpha_i}(n)$ grows as $\mathcal{O}(1/X_{i\alpha_i}(n-1))$, implying that (H2)/(3.2b) may fail to hold if the players' action selection probabilities become arbitrarily small.

Importantly, this can never happen if ($\varepsilon$-Hedge) is run with a strictly positive exploration factor $\varepsilon > 0$. In that case, we can show that the bandit estimator (4.2) satisfies both (H1) and (H2), leading to the following result:

**Theorem 3.** *Let $\Gamma$ be a generic potential game and suppose that Algorithm 1 is run with i) the bandit estimator (4.2); ii) a strictly positive exploration factor $\varepsilon > 0$; and iii) a step-size sequence of the form $\gamma_n \propto 1/n^\beta$ for some $\beta \in (0,1]$. Then:*

1. *$X(n)$ converges (a.s.) to a $\delta$-equilibrium of $\Gamma$ with $\delta \equiv \delta(\varepsilon) \to 0$ as $\varepsilon \to 0$.*

2. *If $\lim_{n\to\infty} X(n)$ is an $\varepsilon$-pure state of the form $x_i^* = \varepsilon\,\mathrm{unif}_i + (1-\varepsilon)e_{\hat{\alpha}_i}$ for some $\hat{\alpha} \in \mathcal{A}$, then $\hat{\alpha}$ is a.s. a strict equilibrium of $\Gamma$ and convergence occurs at a quasi-exponential rate:*

$$X_{i\hat{\alpha}_i}(n) \geq 1 - \varepsilon - be^{-c\sum_{k=1}^{n}\gamma_k} \quad \text{for some positive } b, c > 0. \tag{4.4}$$

*Proof.* Under Algorithm 1, the estimator (4.2) gives

$$\|\hat{v}_i(n)\| = \frac{|\hat{u}_i(n)|}{X_{i\alpha_i(n-1)}(n)} \leq \frac{|u_i(\alpha_i(n); \alpha_{-i}(n))|}{\varepsilon} \leq \frac{u_{\max}}{\varepsilon}, \tag{4.5}$$

where $u_{\max} = \max_{i\in\mathcal{N}}\max_{\alpha_1\in\mathcal{A}_1}\cdots\max_{\alpha_N\in\mathcal{A}_N} u_i(\alpha_1,\ldots,\alpha_N)$ denotes the absolute maximum payoff in $\Gamma$. This implies that (H2) holds true for all $q > 2$, so our claim follows from Theorem 1. $\qquad\square$

Theorem 3 shows that the limit of Algorithm 1 is closer to the Nash set of the game if the exploration factor $\varepsilon$ is taken as small as possible. On the other hand, the crucial limitation of this result is that it does not apply to the case $\varepsilon = 0$ which corresponds to the game's bona fide Nash equilibria. As we discussed above, the reason for this is that the variance of $\hat{v}(n)$ may grow without bound if action choice probabilities become arbitrarily small, in which case the main components of our proof break down.

With this "bias-variance" trade-off in mind, we introduce below a modified version of Algorithm 1 with an "annealing" schedule for the method's exploration factor:

---

**Algorithm 2** Exponential weights with annealing

---

**Require:** step-size sequence $\gamma_n > 0$, vanishing exploration factor $\varepsilon_n > 0$, initial scores $Y_i \in \mathbb{R}^{\mathcal{A}_i}$
1: **for** $n = 1, 2, \ldots$ **do**
2:    **for** every player $i \in \mathcal{N}$ **do**
3:       set mixed strategy: $X_i \leftarrow \varepsilon_n\,\mathrm{unif}_i + (1-\varepsilon_n)\,\Lambda_i(Y_i)$;
4:       choose action $\alpha_i \sim X_i$ and receive payoff $\hat{u}_i \leftarrow u_i(\alpha_i; \alpha_{-i})$;
5:       set $\hat{v}_{i\alpha_i} \leftarrow \hat{u}_i/X_{i\alpha_i}$ and $\hat{v}_{i\beta_i} \leftarrow 0$ for $\beta_i \neq \alpha_i$;
6:       update scores: $Y_i \leftarrow Y_i + \gamma_n\hat{v}_i$;
7:    **end for**
8: **end for**

---

Of course, the convergence of Algorithm 2 depends heavily on the rate at which $\varepsilon_n$ decays to $0$ relative to the algorithm's step-size sequence $\gamma_n$. This can be seen clearly in our next result:

**Theorem 4.** *Let $\Gamma$ be a generic potential game and suppose that Algorithm 1 is run with i) the bandit estimator (4.2); ii) a step-size sequence of the form $\gamma_n \propto 1/n^\beta$ for some $\beta \in (1/2, 1]$; and iii) a decreasing exploration factor $\varepsilon_n \downarrow 0$ such that*

$$\lim_{n\to\infty}\frac{\gamma_n}{\varepsilon_n^2} = 0, \quad \sum_{n=1}^{\infty}\frac{\gamma_n^2}{\varepsilon_n} < \infty, \quad \text{and} \quad \lim_{n\to\infty}\frac{\varepsilon_n - \varepsilon_{n+1}}{\gamma_n^2} = 0. \tag{4.6}$$

*Then, $X(n)$ converges (a.s.) to a Nash equilibrium of $\Gamma$.*

The main challenge in proving Theorem 4 is that, unless the "innovation term" $U_i(n) = \hat{v}_i(n) - v_i(X(n-1))$ has bounded variance, Benaïm's general theory does not imply that $X(n)$ forms an asymptotic pseudotrajectory of the underlying mean dynamics – here, the unperturbed replicator system (RD). Nevertheless, under the summability condition (4.6), it is possible to show that this is the case by using a martingale limit argument based on Burkholder's inequality. Furthermore, under the stated conditions, it is also possible to show that, if $X(n)$ converges, its limit is necessarily a Nash equilibrium of $\Gamma$. Our proof then follows in roughly the same way as in the case of Theorem 1; for the details, we refer the reader to the appendix.

We close this section by noting that the summability condition (4.6) imposes a lower bound on the step-size exponent $\beta$ that is different from the lower bound in Theorem 3. In particular, if $\beta = 1/2$, (4.6) cannot hold for any vanishing sequence of exploration factors $\varepsilon_n \downarrow 0$. Given that the innovation term $U_i$ is bounded, we conjecture that this sufficient condition is not tight and can be relaxed further. We intend to address this issue in future work.

## 5   Conclusion and perspectives

The results of the previous sections show that no-regret learning via exponential weights enjoys appealing convergence properties in generic potential games. Specifically, in the semi-bandit case, the sequence of play converges to a Nash equilibrium with probability 1, and convergence to pure equilibria occurs at a quasi-exponential rate. In the bandit case, the same holds true for $\mathcal{O}(\varepsilon)$-equilibria if the algorithm is run with a positive mixing factor $\varepsilon > 0$; and if the algorithm is run with a decreasing mixing schedule, the sequence of play converges to an actual Nash equilibrium (again, with probability 1). In future work, we intend to examine the algorithm's convergence properties in other classes of games (such as smooth games), extend our analysis to the general "follow the regularized leader" (FTRL) class of policies (of which EW is a special case), and to examine the impact of asynchronicities and delays in the players' feedback/update cycles.

**Acknowledgments**

Johanne Cohen was partially supported by the grant CNRS PEPS MASTODONS project ADOC 2017. Amélie Héliou and Panayotis Mertikopoulos gratefully acknowledge financial support from the Huawei Innovation Research Program ULTRON and the ANR JCJC project ORACLESS (grant no. ANR–16–CE33–0004–01).

## Footnotes

[1]Having a exploration factor $\varepsilon > 0$ simply means here that action selection probabilities never fall below $\varepsilon$.

[2]In the above $(\alpha_i; \alpha_{-i})$ is shorthand for $(\alpha_1, \dots, \alpha_i, \dots, \alpha_N)$, used here to highlight the action of player $i$ against that of all other players.

[3]Of course, the exploration factor $\varepsilon$ could also be player-dependent. For simplicity, we state all our results here with the same $\varepsilon$ for all players.

[4]Notation-wise, this means that the players' actions at stage $n$ are drawn based on their mixed strategies at stage $n-1$. This slight discrepancy with the pseudocode representation of Algorithm 1 is only done to simplify notation later on.

[5]Specifically, we refer here to the so-called "folk theorem" of evolutionary game theory which states that $x^*$ is asymptotically stable under (RD) if and only if it is a strict Nash equilibrium of $\Gamma$ [15]. The extension of this result to the $\varepsilon$-replicator system (RD$_\varepsilon$) is immediate.

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
