[Supplementary Material · Main-supp.pdf]

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

# A    Learning with semi-bandit feedback

This section is devoted to the proof of Theorem 1. In Section 3, the recursion equation of Algorithm 1 depends on $\varepsilon$ parameter. In this section, each player $i$ can have its own mixing factor $\varepsilon_i$. This choice is done in order to simplify the notation in Sections 3 and 4. Thus the recursion equation of Algorithm 1 can be rewritten as

$$
\begin{aligned}
X_i(n) &= \varepsilon_i/|\mathcal{A}_i|\,\mathbf{1} + (1 - \varepsilon_i)\,\Lambda_i(Y_i(n)), \\
Y_i(n+1) &= Y_i(n) + \gamma_{n+1}\hat{v}_i(n+1),
\end{aligned} \qquad (\varepsilon\text{-Hedge})
$$

To perform this, we introduce some other discrete-time process $(X_i^{\Lambda}(n))_{n \in \mathbb{N}}$ to help with this proof.

$$
X_i^{\Lambda}(n) = \frac{1}{(1 - \varepsilon_i)}(X_i(n) - \varepsilon_i/|\mathcal{A}_i|\mathbf{1})
$$

By definition, we have $X_i(n) = \varepsilon_i/|\mathcal{A}_i|\mathbf{1} + (1 - \varepsilon_i)X_i^{\Lambda}(n)$. Besides,

**Observation 5.** *For any $n \in \mathbb{N}$, $\|X(n) - X^{\Lambda}(n)\|_2 \leq N\epsilon$, where $\epsilon = max_{i \in \mathbb{N}} \frac{\epsilon_i}{\sqrt{|\mathcal{A}_i|}}$.*

Moreover, we can establish an other relation between these two discrete-time processes:

**Lemma 6.** *If $(X^{\Lambda}(n))_{n \in \mathbb{N}}$ converges to a Nash equilibrium $x^*$, then $(X(n))_{n \in \mathbb{N}}$, defined in ($\varepsilon$-Hedge), converges to a $\delta(\epsilon)$-Nash equilibrium with $\epsilon = \max_{i \in \mathcal{N}} \frac{\epsilon_i}{|\mathcal{A}_i|}$ and $\delta(\epsilon)$ such that if $\epsilon$ tends to 0, then $\delta(\epsilon)$ also tends to 0.*

*Proof.* If $(X^{\Lambda}(n))_{n \in \mathbb{N}}$ converges to a point $x^*$, then $(X(n))_{n \in \mathbb{N}}$ also converges to a limit $\hat{x}$. Observation 5 gives us that $\|X(n) - X^{\Lambda}(n)\|_2 \leq N\epsilon$, by continuity, $\|x^* - \hat{x}\|_2 \leq N\epsilon$.

It remains to show that if the two mixed strategy profiles $p$ and $p'$ are such that $\|p - p'\|_2 \leq \epsilon$ then $\|u_i(p) - u_i(p')\|_2 \leq \frac{\delta(\epsilon)}{2}$, for any player $i \in \mathcal{N}$.

$$
\begin{aligned}
|u_i(p) - u_i(p')| &= \Big| \sum_{\alpha_1 \in \mathcal{A}_1} \cdots \sum_{\alpha_N \in \mathcal{A}_N} u_i(\alpha_1, \ldots, \alpha_N)(p_{1\alpha_1} \cdots p_{N\alpha_N} - p'_{1\alpha_1} \cdots p'_{N\alpha_N}) \Big| \\
&\leq \sum_{\alpha_1 \in \mathcal{A}_1} \cdots \sum_{\alpha_N \in \mathcal{A}_N} u_i(\alpha_1, \ldots, \alpha_N)|(p_{1\alpha_1} \cdots p_{N\alpha_N} - p'_{1\alpha_1} \cdots p'_{N\alpha_N})| \\
&\leq \sum_{\alpha_1 \in \mathcal{A}_1} \cdots \sum_{\alpha_N \in \mathcal{A}_N} u_i(\alpha_1, \ldots, \alpha_N)|(p_{1\alpha_1} \cdots p_{N\alpha_N} - (p_{1\alpha_1} - \epsilon) \cdots (p_{N\alpha_N} - \epsilon))| \\
&\leq \sum_{\alpha_1 \in \mathcal{A}_1} \cdots \sum_{\alpha_N \in \mathcal{A}_N} u_i(\alpha_1, \ldots, \alpha_N) \sum_{k=1}^{N} (-\epsilon)^k \binom{N}{k} \\
&= \frac{\delta(\epsilon)}{2}
\end{aligned}
$$
$$(A.1)$$

Moreover, observe that if $\epsilon \to 0$, then $\frac{\delta(\epsilon)}{2} \to 0$, and $|u_i(p) - u_i(p')| \to 0$.

Assume that $(X^{\Lambda}(n))_{n \in \mathbb{N}}$ converges to a Nash equilibrium $x^*$. By definition, for all $x_i \in X_i, i \in \mathbb{N}$ $u_i(x_i^*; x_{-i}^*) \geq u_i(x_i; x_{-i}^*)$. In addition, we have $\|x^* - \hat{x}\|_2 \leq N\epsilon$ and $\|(x_i, x_{-i}^*) - (x_i, \hat{x}_{-i})\|_2 \leq N\epsilon \quad \forall x_i \in X_i$.

$$
\begin{aligned}
u_i(\hat{x}_i; \hat{x}_{-i}) + \frac{\delta(N\epsilon)}{2} &\geq u_i(x_i^*; x_{-i}^*) \geq u_i(x_i; x_{-i}^*) \geq u_i(x_i; \hat{x}_{-i}) - \frac{\delta(N\epsilon)}{2} \\
u_i(\hat{x}_i; \hat{x}_{-i}) + \delta(N\epsilon) &\geq u_i(x_i; \hat{x}_{-i}) \text{ for all } x_i^* \in \mathcal{X}_i, i \in \mathcal{N}
\end{aligned}
$$
$$(A.2)$$

So, the latter equation corresponds to the definition of $\delta(\epsilon)$-equilibrium. We can conclude that $(X(n))_{n \in \mathbb{N}}$ converges to a $\delta(\epsilon)$-Nash equilibrium with $\delta(\epsilon)$. $\qquad \square$

The remainder of this section is devoted to prove the convergence results of $\varepsilon$-HEDGE Algorithm, and it is split into two parts according to the feedback assumption. The proof is based on Benaim's study of stochastic approximations [28]. We follow 4 points:

1. Show that $X$ is an asymptotic pseudo trajectory of a continuous dynamics,

2. Show that the potential function of the game is strict Lyapunov function of the dynamics,

3. Show that $X$ converges toward a rest point of the dynamics,

4. Show that if $X$ converges toward a point it is a Nash Equilibrium.

For the first step, we focus on the sequences $(X_i(n))_{n \in \mathbb{N}}$ and its linear interpolation $(x_i(t))$. We also define the continuous variables, $x_{i\alpha}^{\Lambda}(t) = \frac{x_{i\alpha}(t) - \epsilon_i / |\mathcal{A}|_i}{1 - \epsilon_i}$ and $y_{i\alpha}(t)$ such that $x_{i\alpha}^{\Lambda}(t) = \Lambda_i(y_{i\alpha}(t))$, with the same relations as the corresponding discret processes. We will prove that the sequences $(X_i(n))_{n \in \mathbb{N}}$ correspond to some classical family of stochastic approximation algorithms so-called *Approximate Robbins-Monro algorithm*.

**Definition 7** (Approximate Robbins-Monro conditions)**.** The general stochastic approximation algorithm $x(n+1) = x(n) + \gamma_n(F(x(n)) + U_n + \beta_n)$ is said to be an *approximate Robbins-Monro algorithm* if:

- $F : \mathbb{R}^m \to \mathbb{R}^m$ is a continuous map;

- $U_n \in \mathbb{R}^m$ are perturbations and $U_n$ is a martingale difference noise;

- $\{\gamma_n\}_{n \geq 1}$ is a given sequence of nonnegative numbers such that $\sum_k \gamma_k = \infty$ and $\lim_{n \to \infty} \gamma_n = 0$;

- $\lim_{n \to \infty} b_n = 0$ almost surely.

Then, we will prove that the interpolated process of the sequences $(X_i(n))_{n \in \mathbb{N}}$ is an asymptotic pseudo trajectory of the solutions of the following ordinary differential equation.

$$
\begin{aligned}
\dot{x}_{i\alpha} &= \left(x_{i\alpha} - |\mathcal{A}_i|^{-1} \varepsilon_i\right)\left[v_{i\alpha}(x) - (1 - \varepsilon_i)^{-1} \sum_{\beta \in \mathcal{A}_i} (x_{i\beta} - |\mathcal{A}_i|^{-1} \varepsilon_i) v_{i\beta}(x)\right] \\
&= \left(1 - \varepsilon_i\right) x_{i\alpha}^{\Lambda}\left[v_{i\alpha}(x) - \sum_{\beta \in \mathcal{A}_i} x_{i\beta}^{\Lambda} v_{i\beta}(x)\right]
\end{aligned}
\tag{A.3}
$$

To proceed, recall the definition of the asymptotic pseudo-trajectory.

**Definition 8** (Asymptotic Pseudo-trajectories)**.** Given a flow $\phi : \mathbb{R} \times M \to M, (n, x) \to \phi(n, x) = \phi_n(x)$ such that $\phi_0 = $ Identity and $\phi_{n+\alpha} = \phi_n \circ \phi_\alpha$, a continuous function $X : \mathbb{R} \to M$ is an asymptotic pseudo-trajectory if

$$
\lim_{n \to \infty} \sup_{0 \leq k \leq T} d((X(n+k), \phi_h(X(n))) = 0 \text{ for any } T > 0
\tag{A.4}
$$

**Proposition 9.** *Suppose that Algorithm 1 is run with i) semi-bandit feedback satisfying (H1) and (H2); ii) a nonnegative mixing factor $\varepsilon \geq 0$; and iii) a step-size sequence of the form $\gamma_n \propto 1/n^\beta$ for some $\beta \in (1/q, 1]$. The interpolated process of the sequences $(X_i(n))_{n \in \mathcal{N}}$ is an asymptotic pseudo trajectory of the solutions of the following ordinary differential equation*

$$
\dot{x}_{i\alpha} = \left(x_{i\alpha} - |\mathcal{A}_i|^{-1} \varepsilon_i\right)\left[v_{i\alpha}(x) - (1 - \varepsilon_i)^{-1} \sum_{\beta \in \mathcal{A}_i} (x_{i\beta} - |\mathcal{A}_i|^{-1} \varepsilon_i) v_{i\beta}(x)\right],
\tag{$\text{RD}_\varepsilon$}
$$

*Proof.* The proof of Proposition 9 can be split into two parts. First, we will prove that the stochastic process $\{X_i(n)\}_{n \in \mathbb{N}}$ given by Algorithm 1 is an approximate Robbins-Monro algorithm. Second, we will conclude by applying some classical results in [28].

Observe that for any $i \in \mathcal{N}$, for any $\alpha, \beta, \beta' \in \mathcal{A}_i$, $x_{i\alpha}^{\Lambda} = \Lambda_{i\alpha}(y_i) = \frac{\exp(y_{i\alpha})}{\sum_{s \in \mathcal{A}_i} \exp(y_{i\alpha})}$, we have

$$
\frac{\partial \Lambda_{i\alpha}(y_i)}{\partial y_{i\beta}} = x_{i\alpha}^{\Lambda}(\mathbb{1}_{s=\beta} - x_{i\beta}^{\Lambda}), \text{ and } \frac{\partial^2 \Lambda_{i\alpha}(y_i)}{\partial y_{i\beta} \partial y_{i\beta'}} = x_{i\alpha}^{\Lambda}\left(\mathbb{1}_{\alpha=\beta=\beta'} - \mathbb{1}_{\alpha=\beta} x_{i\beta'}^{\Lambda} - x_{i\beta}^{\Lambda}(\mathbb{1}_{\alpha=\beta'} + \mathbb{1}_{\beta=\beta'} - 2x_{i\beta'}^{\Lambda})\right).
$$

Using Taylor's Remainder Theorem, let us rewrite the equation $X_{i\alpha}(n+1) = \frac{\epsilon_i}{|\mathcal{A}_i|} + (1 - \epsilon_i)\Lambda_{i\alpha}(Y_i(n+1))$ as

$$
\begin{aligned}
X_{i\alpha}(n+1) &= \frac{\epsilon_i}{|\mathcal{A}_i|} + (1 - \epsilon_i)\Lambda_{i\alpha}(Y_i(n) + \gamma_{n+1}\hat{v}_i(n+1)) \\
&= \frac{\epsilon_i}{|\mathcal{A}_i|} + (1 - \epsilon_i)\Lambda_{i\alpha}(Y_i(n)) \\
&\quad + (1 - \epsilon_i)\gamma_{n+1}\left(\nabla\Lambda_{i\alpha}^{\mathrm{T}}(Y_i(n))\hat{v}_i(n+1) + \frac{1}{2}\gamma_{n+1}\hat{v}_i^{\mathrm{T}}(n+1)\operatorname{Hess}\Lambda_{i\alpha}(\psi_i(n))\hat{v}_i(n+1)\right) \\
&= X_{i\alpha}(n) + (1 - \epsilon_i)\gamma_{n+1}\left(\nabla\Lambda_{i\alpha}^{\mathrm{T}}(Y_i(n))\hat{v}_i(n+1) + \frac{\gamma_{n+1}}{2}\hat{v}_i^{\mathrm{T}}(n+1)\operatorname{Hess}\Lambda_{i\alpha}(\psi_i(n))\hat{v}_i(n+1)\right) \\
&= X_{i\alpha}(n) + (1 - \epsilon_i)\gamma_{n+1}\left(\nabla\Lambda_{i\alpha}^{\mathrm{T}}(Y_i(n))v_i(X(n)) + \nabla\Lambda_{i\alpha}^{\mathrm{T}}(Y_i(n))(\hat{v}_i(n+1) - v_i(X(n))) + \gamma_{n+1}a_{n+1}\right)
\end{aligned}
$$
(A.5)

where $\nabla\Lambda_{i\alpha}$ is the gradient vector of $\Lambda_{i\alpha}$, $\nabla\Lambda_{i\alpha}^{\mathrm{T}}$ is its transposed, $\operatorname{Hess}\Lambda_{i\alpha}$ is the Hessian matrix of $\Lambda_{i\alpha}$, $\psi_i(n)$ is in the line segment going out from $Y_i(n)$ to the point $Y_i(n+1)$, and $a_{n+1} = \frac{1}{2}\hat{v}_i^{\mathrm{T}}(n+1)Hess\Lambda_{i\alpha}(\psi_i(n))\hat{v}_i(n+1)$.

Next, we focus on $\gamma_n a_n$ (corresponding to parameter $b_n$ in Definition 7). Since $\frac{\partial^2 \Lambda_{i\alpha}(y_i)}{\partial y_{i\beta}\partial y_{i\beta'}} = x_{i\alpha}^\Lambda\left(\mathbb{1}_{\alpha=\beta=\beta'} - \mathbb{1}_{\alpha=\beta}x_{i\beta'}^\Lambda - x_{i\beta}^\Lambda(\mathbb{1}_{\alpha=\beta'} + \mathbb{1}_{\beta=\beta'} - 2x_{i\beta'}^\Lambda)\right)$, all components of $Hess\Lambda_{i\alpha}(\psi_i(n))$ are bounded. So, the limit of $\gamma_n a_n$ (when $n \to \infty$) depends on the limit of $||\hat{v}_{i\alpha}(n)||^2$.

Let $E_{i\alpha,n}$ be the event $||\hat{v}_{i\alpha}(n)||^2 \geq n^\alpha$ for $\frac{1}{q} < \alpha < \frac{1}{p}$, with $q$ defined in Hypothesis (H2) and $p$ such that $\gamma_n = \mathcal{O}(1/n^{\frac{1}{p}})$.

Hypothesis (H2) gives us that

$$
\sum_{n=0}^{\infty}\mathbb{P}(E_{i\alpha,n}) = \sum_{n=0}^{\infty}\mathbb{P}(||\hat{v}_{i\alpha}(n)||^2 \geq n^\alpha \mid \mathcal{F}_{n-1}) = \sum_{n=0}^{\infty}\mathcal{O}(\frac{1}{n^{q\alpha}}) < \infty
$$
(A.6)

The Borel-Cantelli lemma gives us that $E_{i\alpha,n}$ is true for only a finite number of $n \in \mathbb{N}$. Therefore for $n > max_{m\in\mathbb{N}}\{m; \exists i \in \mathcal{N}, \alpha \in \mathcal{A}_i, E_{i\alpha,m} \text{ is true}\}$, $||\hat{v}_{i\alpha}(n)||^2 < n^\alpha$. By assumption, $\gamma_n = o(n^b)$ for any $b > -1/p$. In particular, $\gamma_n = o(n^{-\alpha})$ so $\lim_{n\to\infty} a_n\gamma_n = 0$.

$$
X_{i\alpha}(n+1) = X_{i\alpha}(n) + (1 - \epsilon_i)\gamma_{n+1}\left(X_{i\alpha}^\Lambda(n)\left(v_{i\alpha}(X(n)) - \sum_{\beta\in\mathcal{A}_i}v_{i\beta}(X(n))X_{i\beta}^\Lambda(n)\right)\right)
$$

$$
+ (1 - \epsilon_i)\gamma_{n+1}\left(X_{i\alpha}^\Lambda(n)(\hat{v}_{i\alpha}(n+1) - v_{i\alpha}(X(n)) - \sum_{\beta\in\mathcal{A}_i}X_{i\beta}^\Lambda(n)[\hat{v}_{i\beta}(n+1) - v_{i\beta}(X(n))]) + \gamma_n a_n\right)
$$

Let $U_{i\alpha,n} = (1 - \epsilon_i)X_{i\alpha}^\Lambda(n)(\hat{v}_{i\alpha}(n+1) - v_{i\alpha}(X(n)) - \sum_{\beta\in\mathcal{A}_i}X_{i\beta}^\Lambda(n)[\hat{v}_{i\beta}(n+1) - v_{i\beta}(X(n))])$.

Recall that (3.2a) and (3.2b) can be deduced from Hypotheses (H1) (H2). We get

1. $\mathbb{E}[U_{i\alpha,n}|\mathcal{F}_{n-1}] = 0$ for all $n$

2. $\mathbb{E}[||U_{i\alpha,n}||^2] < \infty$ for all $n$

So, $U_{i\alpha,n}$ is a martingale difference noise.

Since the function which to $x_{i\alpha}$ associates $(1 - \varepsilon)x_{i\alpha}^\Lambda\left[v_{i\alpha}(x) - \sum_{\beta\in\mathcal{A}_i}x_{i\beta}^\Lambda v_{i\beta}(x)\right]$ is a continuous map, we can conclude that the stochastic process $\{X_i(n)\}_{n\in\mathbb{N}}$ is an approximate Robbins-Monro algorithm.

Second, Remark 4.5 and Propositions 4.2 and 4.1 of [28] allow us to conclude that the interpolated process of the sequences $(X_i(n))_{n\in\mathbb{N}}$ is an asymptotic pseudo trajectory of the solutions of ODE $(\text{RD}_\varepsilon)$.

$\square$

Dynamics $(\text{RD}_\varepsilon)$ can be viewed as a $\varepsilon$-perturbed variant of the replicator dynamics. that readily extends to the $\varepsilon$-perturbed variant $(\text{RD}_\varepsilon)$. Moreover, the replicator dynamics have some good property in potential games. The next theorem expresses this property in our context:

**Proposition 10.** *Let $\Gamma$ be a generic potential game. The potential function $f$ of $\Gamma$ is a strict increasing Lyapunov function of the flow inducted by the dynamics $(\text{RD}_\varepsilon)$.*

*Proof.* We consider the variation of $f$. We have

$$\dot{f}(x) = \sum_{i\in\mathcal{N}}\sum_{\alpha\in\mathcal{A}_i} \frac{\partial f}{\partial x_{i\alpha}}(x)\dot{x}_{i\alpha}$$

$$= \sum_{i\in\mathbb{N}} v_i^{\mathrm{T}}(x(t))\dot{x}_i(t)$$

$$= \sum_{i\in\mathbb{N}}\sum_{\alpha\in\mathcal{A}_i} v_{i\alpha}(x(t))\dot{x}_{i\alpha}(t)$$

$$= \sum_{i\in\mathbb{N}}\sum_{\alpha\in\mathcal{A}_i} (1-\varepsilon_i)v_{i\alpha}(x(t))x_{i\alpha}^\Lambda(t)\left(v_{i\alpha}(x(t)) - \sum_{\beta\in\mathcal{A}_i} v_{i\beta}(x(t))x_{i\beta}^\Lambda(t)\right)$$

$$= \sum_{i\in\mathbb{N}}\sum_{\alpha\in\mathcal{A}_i}\sum_{\beta\in\mathcal{A}_i,\beta>\alpha} (1-\varepsilon_i)x_{i\alpha}^\Lambda(t)x_{i\beta}^\Lambda(t)[v_{i\alpha}(x(t)) - v_{i\beta}(x(t))]^2$$

The second equation is obtained by definitions of potential game and dynamic $(\text{RD}_\varepsilon)$ (definition of $\dot{x}_{i\alpha}$). From the latter equation, we can conclude that $\dot{f}(x) \geq 0$.

Now we show that the rest points $x$ of dynamics $(\text{RD}_\varepsilon)$ are such that $\dot{f}(x) = 0$. Since $\dot{f}(x) = \sum_{i\in\mathcal{N}}\sum_{\alpha\in\mathcal{A}_i} \frac{\partial f}{\partial x_{i\alpha}}(x)\dot{x}_{i\alpha}$, $\dot{f}(x) = 0$ when $x$ is a rest point ($\dot{x} = 0$ ).

Conversely, we will prove that $\dot{f}(x) = 0$ implies that $x$ is a rest point of dynamics $(\text{RD}_\varepsilon)$.

Observe that $\dot{f}(x) = 0$ implies that for all players $i$ in $\mathcal{N}$, and pure strategies $\alpha$ and $\beta$ in $\mathcal{A}_i$ we have $x_{i\alpha}^\Lambda = 0$ or $x_{i\beta}^\Lambda = 0$ or $v_{i\alpha}(x) = v_{i\beta}(x)$.

Since the dynamics $(\text{RD}_\varepsilon)$ is $\dot{x}_{i\alpha} = (1-\varepsilon_i)x_{i\alpha}^\Lambda\left(v_{i\alpha}(x) - \sum_{\beta\in\mathcal{A}_i} v_{i\beta}(x)x_{i\beta}^\Lambda\right)$, if for all players $i$ in $\mathcal{N}$, and pure strategies $\alpha$ and $\beta$ in $\mathcal{A}_i$ we have $x_{i\alpha}^\Lambda = 0$ or $x_{i\beta}^\Lambda = 0$, then we can deduce $\dot{x}_{i\alpha} = 0$. Otherwise, if $v_{i\alpha}(x) = v_{i\beta}(x)$ for all $\alpha$ and $\beta$ in $\mathcal{A}_i$ such that $x_{i\alpha}^\Lambda \neq 0$ and $x_{i\beta}^\Lambda \neq 0$, we have then $\dot{x}_{i\alpha} = 0$

To conclude, $f$ is increasing, and its derivative is null if and only if it is evaluated on a rest point of the dynamics $(\text{RD}_\varepsilon)$. $f$ is a strict increasing Lyapunov function of the dynamics $(\text{RD}_\varepsilon)$. $\square$

**Proposition 11.** *Let $\Gamma$ be a generic potential game. Suppose that Algorithm 1 is run with i) semi-bandit feedback satisfying (H1) and (H2); ii) a nonnegative mixing factor $\varepsilon \geq 0$; and iii) a step-size sequence of the form $\gamma_n \propto 1/n^\beta$ for some $\beta \in (1/q, 1]$. The interpolated process of the sequences $(X_i(n))_{n\in\mathbb{N}}$ converges to a rest point of $\text{RD}_\varepsilon$.*

*Proof.* We showed that under theses assumptions, $(X_i(n))_{n\in\mathbb{N}}$ is a pseudo asymptotic trajectory of the flow induced by the dynamics $\text{RD}_\varepsilon$.

We now show that $\text{RD}_\varepsilon$ has a finite number of rest points. When $\forall i \in \mathcal{N}, \varepsilon_i = 0$, we have $x^\Lambda = x$. Thus, we obtain from the previous proof that $x$ is a rest point of $\text{RD}_\varepsilon$ if and only if

$$\sum_{i\in\mathbb{N}}\sum_{\alpha\in\mathcal{A}_i}\sum_{\beta\in\mathcal{A}_i,\beta>\alpha} (1-\varepsilon_i)x_{i\alpha}(t)x_{i\beta}(t)[v_{i\alpha}(x(t)) - v_{i\beta}(x(t))]^2 = 0 \qquad (\text{A.7})$$

So $x$ is a rest point of $RD_\varepsilon$ if and only if $v_{i\alpha}(x(t)) = v_{i\beta}(x(t))$ $\forall \alpha, \beta \in supp(x_i)$. Therefore $x$ is a rest point of $RD_\varepsilon$ if and only if it is a restricted equlibrium. The game is finite so they are a finite number of game restrictions. Each restricted game is a finite, generic, potential game so it has a finite number of Nash-equilibrium.

Now we address the other case : $\exists i \in \mathcal{N}, \varepsilon_i \neq 0$. Thus, $x$ is a rest point of $RD_\varepsilon$ if and only if $v_{i\alpha}(x(t)) = v_{i\beta}(x(t))$ $\forall \alpha, \beta \in supp(x_i^\Lambda)$. Let $supp_i$ be the cardinal of $supp(x_i^\Lambda)$. For each player $i$, we have $supp_i - 1$ unknown coordinates corresponding to the $x_i^\Lambda$ non-zero coordinates. Thus, $supp_i - 1$ independent equations corresponding to $v_{i\alpha}(x(t)) = v_{i\beta}(x(t))$ $\forall \alpha, \beta \in supp(x_i^\Lambda)$ because the game is generic. Such a system has only an unique solution. Therefore for each possible support of $x$ there is only one rest point, and dynamics $RD_\varepsilon$ has a finite number of rest points.

Corollary 6.6 of [28] allows to conclude that the continous-time process $x_i(t)$ converges to a rest point of $RD_\varepsilon$. □

In order to prove Point 1 of Theorem 1, we need to have the following technical result. This lemma is about the properties on rationality properties (such as comparaisons among strategies corresponding to the elimination of dominated strategies).

**Lemma 12.** *Suppose that Algorithm 1 is run with i*) *semi-bandit feedback satisfying* (H1) *and* (H2); *ii*) *a nonnegative mixing factor $\varepsilon \geq 0$; and iii*) *a step-size sequence of the form $\gamma_n \propto 1/n^\beta$ for some $\beta \in (1/q, 1]$. If there exists some $a > 0$ such that $v_\beta(x) - v_\alpha(x) \geq a$ for all $x \in \mathcal{X}$ then for all $c \in (0, a)$, there exists some $n_0$ such that $Y_\beta(n) - Y_\alpha(n) \geq c \sum_{k=1}^n \gamma_k$ for all $n \geq n_0$ (a.s.).*

*Proof.* Let $\zeta_k = \hat{v}_\beta(k) - v_\beta(X(k-1)) - [\hat{v}_\alpha(k) - v_\alpha(X(k-1))]$. By assumption there exists $a > 0$ such that $v_\beta(X) - v_\alpha(X) \geq a$ for all $X \in \mathcal{X}$. Then,

$$
\begin{aligned}
Y_\beta(n) - Y_\alpha(n) &= Y_\beta(0) - Y_\alpha(0) + \sum_{k=1}^n \gamma_k \left( \hat{v}_\beta(k) - \hat{v}_\alpha(k) \right) \\
&= Y_\beta(0) - Y_\alpha(0) + \sum_{k=1}^n \gamma_k \left[ v_\beta(X(k-1)) - v_\alpha(X(k-1)) \right] + \sum_{k=1}^n \gamma_k \zeta_k \quad \text{(A.8)} \\
&\geq Y_\beta(0) - Y_\alpha(0) + \sum_{k=1}^n \gamma_k \left[ a + \frac{\sum_{k=1}^n \gamma_k \zeta_k}{\sum_{k=1}^n \gamma_k} \right].
\end{aligned}
$$

Now we will prove that $\frac{\sum_{k=1}^n \gamma_k \zeta_k}{\sum_{k=1}^n \gamma_k} \to 0$.

The reformulation of Hypothesis (H1) gives :

$$
\begin{aligned}
\mathbb{E}[\zeta_k | \mathcal{F}_{k-1}] &= \mathbb{E}[\xi_\beta(k) + v_\beta(s(k)) - v_\beta(X(k-1)) - \xi_\alpha(k) - v_\alpha(\alpha(k)) + v_\alpha(X(k-1)) | \mathcal{F}_{k-1}] \\
&= \mathbb{E}[\xi_\beta(k) - \xi_\alpha(k) | \mathcal{F}_{k-1}] + \mathbb{E}[v_\beta(\alpha(k)) | \mathcal{F}_{k-1}] - v_\beta(X(k-1)) - \mathbb{E}[v_\alpha(\alpha(k)) | \mathcal{F}_{k-1}] + v_\alpha(X(k-1)) \\
&= 0
\end{aligned}
$$
(A.9)

$\zeta_k$ is $\mathcal{F}_k$-measurable, meaning that it is fully determined by the information of $\mathcal{F}_k$.

With $S_n = \sum_{k=1}^n \gamma_k \zeta_k$, it follows that

$$
\mathbb{E}[S_k | \mathcal{F}_{k-1}] = \gamma_k \mathbb{E}[\zeta_k | \mathcal{F}_{k-1}] + \mathbb{E}[S_{k-1} | \mathcal{F}_{k-1}] = S_{k-1} \quad \text{(A.10)}
$$

Therefore $\{S_n = \sum_{k=1}^n \gamma_k \zeta_k, \mathcal{F}_n, n \geq 1\}$ is a martingale, in addition as $\gamma_t$ depends only on $n$ and it is positive, $U_n = \sum_{k=1}^n \gamma_k$ is a nondecreasing sequence of positive random variable such that $U_n$ is $\mathcal{F}_{n-1}$-measurable for each n. In addition $\beta \leq 1$ gives us that $lim_{n \to \infty} U_n = \infty$.

We focus now on proving the last hypothesis of Theorem 2.18 of [31], i.e., $\sum_{k=1}^\infty \frac{\mathbb{E}(\|\gamma_k \zeta_k\|^2 | \mathcal{F}_{k-1})}{U_k^2} < \infty$. First we show that $\mathbb{E}[\|\zeta(k)\|^2 | \mathcal{F}_{k-1}] \leq 4\sigma^2$:

$$
\begin{aligned}
\mathbb{E}[\|\zeta(k)\|^2 | \mathcal{F}_{k-1}] &= \mathbb{E}[\|\hat{v}_\beta(k) - v_\beta(X(k-1)) - [\hat{v}_\alpha(k) - v_\alpha(X(k-1))]\|^2 | \mathcal{F}_{k-1}] \\
&= 2\mathbb{E}[\|\hat{v}_\beta(k) - v_\beta(X(k-1))\|^2 | \mathcal{F}_{k-1}] + 2\mathbb{E}[\|\hat{v}_\alpha(k) - v_\alpha(X(k-1))\|^2 | \mathcal{F}_{k-1}] \\
&\leq 4\sigma^2
\end{aligned}
$$
(A.11)

according to Equation (3.2b). Second, $\gamma_t$ is decreasing so $U_n = \sum_{k=1}^{n} \gamma_k \geq n\gamma_n$ and $U_n^{-2} \leq \frac{1}{n^2\gamma_n^2}$.
Therefore

$$\sum_{k=1}^{\infty} \frac{\mathbb{E}(\|\gamma_k \zeta_k\|^2 | \mathcal{F}_{k-1})}{U_k^2} < \sum_{k=1}^{\infty} \frac{\gamma_k^2 4\sigma^2}{k^2 \gamma_k^2} = 4\sigma^2 \sum_{k=1}^{\infty} \frac{1}{k^2} < \infty \qquad (\text{A.12})$$

Therefore all hypotheses are fulfilled and

$$\lim_{n \to \infty} \frac{\sum_{k=1}^{n} \gamma_k \zeta_k}{\sum_{k=1}^{n} \gamma_k} = 0 \quad (a.s.) \qquad (\text{A.13})$$

So for all $c \in (0, a)$ there exist some $t_0$ such that $-\frac{\sum_{k=1}^{n} \gamma_k \zeta_k}{\sum_{k=1}^{n} \gamma_k} \leq \frac{Y_\beta(0) - Y_\alpha(0)}{\sum_{k=1}^{n} \gamma_k} + a - c$ for all $t > t_0$.
Putting that in (A.8) we have :

$$Y_\beta(n) - Y_\alpha(n) \geq c \sum_{k=1}^{n} \gamma_k \quad \text{for all } t \geq t_0 \text{ (a.s.)} \qquad (\text{A.14})$$
$\square$

Now, we will prove the convergence of the discrete-time process $(X_i(n))_{n \in \mathbb{N}}$.

**Theorem 1** (*Part 1*). *Let $\Gamma$ be a generic potential game. Suppose that Algorithm 1 is run with i) semi-bandit feedback satisfying (H1) and (H2); ii) a nonnegative mixing factor $\varepsilon \geq 0$; and iii) a step-size sequence of the form $\gamma_n \propto 1/n^\beta$ for some $\beta \in (1/q, 1]$. $X(n)$ converges (a.s.) to a $\delta$-equilibrium of $\Gamma$ with $\delta \equiv \delta(\varepsilon) \to 0$ as $\varepsilon \to 0$.*

*Proof.* When $X(n)$ converges, $X^\Lambda(n)$ also converges (by definition). Applying Lemma 6, it suffices to prove that if $X^\Lambda(n)$ converges to $x^*$ then $x^*$ is Nash Equilibrium. We prove by contradiction that $(X^\Lambda(n))_{n \in \mathbb{N}}$ converges to $x^*$ a Nash Equilibrium. Assume that $x^*$ is not a Nash Equilibrium. By definition, we have

$$\exists i \in \mathcal{N}, \exists \beta \in \mathcal{A}_i, \beta \notin supp(x_i^*), s.t., v_{i\beta}(x^*) > v_{i\alpha}(x^*), \forall s \in supp(x_i^*).$$

By continuity of utility $u$, there is a neighborhood $U$ of $x^*$ and $a > 0$ such that:

$\exists i \in \mathcal{N}, \exists \beta \in \mathcal{A}_i, \beta \notin supp(x_i^*), s.t., v_{i\beta}(X^\Lambda) - v_{i\alpha}(X^\Lambda) > a, \forall \alpha \in supp(x_i^*), X^\Lambda \in U$ For $\varepsilon$ small enough and for all $n$ big enough, $X(n) \in U$ because $\|X(n) - x^*\| \leq \|X(n) - X^\Lambda(n)\| + \|X^\Lambda(n) - x^*\| \leq N\varepsilon + \|X^\Lambda(n) - x^*\|$. So, $\exists i \in \mathcal{N}, \exists \beta \in \mathcal{A}_i, \beta \notin supp(x_i^*), s.t., v_{i\beta}(X(n)) - v_{i\alpha}(X(n)) > a, \forall s \in supp(x_i^*)$ Using Lemma 12, for $n_0$ big enough and $n \geq n_0$:

$$Y_{i\beta}(n) - Y_{i\alpha}(n) \geq C + b \sum_{t=n_0}^{n} \gamma_n \to \infty \qquad (\text{A.15})$$

Thus $\frac{X_{i\beta}(n)}{X_{i\alpha}(n)} = \exp\left(Y_{i\beta}(n) - Y_{i\alpha}(n)\right) \to \infty$.

$X_{i\alpha}(n) \to 0$ and $\alpha$ is not in the support of $x_i^*$ which is a contradiction. Therefore $x^*$ is a Nash Equilibrium and $x^*$ is a $\delta(\varepsilon)$-Nash Equilibrium. $\square$

Finally, we will prove the convergence rate of the discrete-time process $(X_i(n))_{n \in \mathbb{N}}$ corresponding to Part 2 of Theorem 1.

**Theorem 1** (*Part 2*). *Let $\Gamma$ be a generic potential game and suppose that Algorithm 1 is run with i) semi-bandit feedback satisfying (H1) and (H2); ii) a nonnegative mixing factor $\varepsilon \geq 0$; and iii) a step-size sequence of the form $\gamma_n \propto 1/n^\beta$ for some $\beta \in (1/q, 1]$. . Then If $\lim_{n \to \infty} X(n)$ is an $\varepsilon$-pure state of the form $x_i^* = \varepsilon/|\mathcal{A}_i|\mathbf{1} + (1-\varepsilon)e_{\hat{\alpha}_i}$ for some $\hat{\alpha} \in \mathcal{A}$, then $\hat{\alpha}$ is a.s. a strict equilibrium of $\Gamma$ and convergence occurs at a quasi-exponential rate:*

$$X_{i\hat{\alpha}_i}(n) \geq 1 - \varepsilon - be^{-c\sum_{k=1}^{n} \gamma_k} \quad \text{for some positive } b, c > 0. \qquad (4.4)$$

*Proof.* We will focus on the sequence $(X_i^\Lambda(n))_{n \in \mathbb{N}}$ and its limit $x^*$. From the proof of Theorem 1 (Part 1), $x^*$ is a Nash equilibrium. By continuity of $u$, there is a neighborhood $U$ of $x^*$ and $a' > 0$ such that:

$\forall i \in \mathcal{N}, \forall \beta \in \mathcal{A}_i, \beta \neq \alpha_i^*, v_{i\alpha_i^*}(x) - v_{i\beta}(x) > a', x \in U.$

Therefore, for $\epsilon$ small enough and for all $n$ big enough, $X(n) \in U$ because $\|X(n) - x^*\| \leq \|X(n) - X^\Lambda(n)\| + \|X^\Lambda(n) - x^*\| \leq N\epsilon + \|X(n) - x^*\|$. So, $\exists i \in \mathbb{N}, \exists \beta \in \mathcal{A}_i, \beta \notin supp(x_i^*), s.t., v_{i\beta}(X^\Lambda(n)) - v_{i\alpha}(X^\Lambda(n)) > a, \forall \alpha \in supp(x_i^*)$ Using Lemma 12 for $n_0$ big enough:

$$Y_{i\alpha^*}(n) - Y_{i\alpha}(n) \geq C + b \sum_{k=n_0}^{n} \gamma_k$$

So, by computation, we can deduce that

$$\sum_{s \in \mathcal{A}_i, \alpha \neq \hat{\alpha}} \exp(Y_{i\alpha}(n) - Y_{i\hat{\alpha}}(n)) \leq \sum_{s \in \mathcal{A}_i, \alpha \neq \hat{\alpha}} \exp(-C_{n_0} - \sum_{k=n_0}^{n} \gamma_k a) \leq C \exp(-\sum_{k=n_0}^{n} \gamma_k a)$$

where we set $C = |\mathcal{A}_i| \exp(-C_{t_0})$. Now, we will focus on $x_{\hat{\alpha}}^\Lambda(n)$.

$$\begin{aligned} X_{i\hat{\alpha}}^\Lambda(n) &= \frac{\exp(Y_{i\hat{\alpha}}(n))}{\sum\limits_{s \in \mathcal{A}} \exp(Y_{i\alpha}(n))} = \frac{1}{\sum\limits_{s \in \mathcal{A}_i} \exp(Y_{i\alpha}(n) - Y_{i\hat{\alpha}}(n))} \\ &= \frac{1}{1 + \sum\limits_{s \in \mathcal{A}_i, s \neq \hat{\alpha}} \exp(Y_{i\alpha}(n) - Y_{i\hat{\alpha}}(n))} \\ &\geq \frac{1}{1 + C \exp(-\sum_{k=n_0}^{n} \gamma_k a)} \end{aligned}$$

The first equation is due to the relation $\frac{\exp a}{\exp b} = \frac{1}{\exp a - b}$. The second equation is obtained by the fact that $\exp(\exp(Y_{i\hat{\alpha}}(n) - Y_{i\hat{\alpha}}(n))) = 1$. Since for any $z > 0$, $\frac{1}{1+z} \geq 1 - z$, we obtain $1 - X_{i\hat{\alpha}}^\Lambda(n) \leq C \exp(-\sum_{k=n_0}^{n} \gamma_k a)$. And, the theorem holds since $|X_{i\hat{\alpha}}^\Lambda(n) - X_{i\hat{\alpha}}(n)| \leq \varepsilon$. $\qquad \square$

## B  Learning with bandit feedback

We begin our proof of Theorem 4 with some general properties of approximate Robbins–Monro algorithms. To state them, let $F : \mathbb{R}^m \to \mathbb{R}^m$ be a continuous map and consider a Robbins-Monro algorithm of the form

$$x_{n+1} - x_n = \gamma_{n+1} \left( F(x_n) + U_{n+1} + b_{n+1} \right) \tag{B.1}$$

where

- $\{\gamma_n\}_{n \geq 1}$ is a given sequence of nonnegative numbers such that $\sum_{n=1}^{\infty} \gamma_n = \infty, \lim_{n \to \infty} \gamma_n = 0$,

- $U_n \in \mathbb{R}^m$ are martingale difference perturbations ,i.e., $\mathbb{E}[U_{n+1} \mid \mathcal{F}_n] = 0$ and $\mathbb{E}\left[\|U_n\|^2\right] < \infty \forall n$,

- $b_n \in \mathbb{R}^m$ is an error term such that $\lim_{n \to \infty} b_n = 0$ with probability 1.

We define:

- $\tau_0 = 0$ and $\tau_n = \sum_{i=1}^{n} \gamma_i$ for $n \geq 1$,
- $m : \mathbb{R}_+ \to \mathbb{N}$ by $m(t) = \sup\{k \geq 0 : t \geq \tau_k\}$,
- $\overline{U} : \mathbb{R}_+ \to \mathbb{N}$ by $\overline{U}(\tau_n + s) = U_{n+1}$, for all $0 \geq s > \gamma_{n+1}$,
- $\overline{\gamma} : \mathbb{R}_+ \to \mathbb{N}$ by $\overline{\gamma}(\tau_n + s) = \gamma_{n+1}$, for all $0 \geq s > \gamma_{n+1}$,

In order to prove that such an algorithm is an asymptotic pseudo trajectories of a given flow it is important to prove that for any $T > 0$, $\lim_{t \to \infty} \sup_{m(t) < h \leq m(t+T)} \|\sum_{i=m(t)}^{h-1} \gamma_{i+1} (U_{i+1} + b_{i+1})\| = 0$ with probability 1.

**Lemma 13.** *Let $\{x_n\}$ be a sequence given by (B.1). Suppose that:*

$$\sum_n \gamma_{n+1}^{\left(1 + \frac{q}{2}\right)} \mathbb{E}[\|U_{n+1}\|^q] < \infty \text{ for a } q \geq 2$$

*Then, for any $T > 0$, $\lim_{t \to \infty} \sup_{m(t) < h \leq m(t+T)} \|\sum_{i=m(t)}^{h-1} \gamma_{i+1} (U_{i+1} + b_{i+1})\| = 0$ with probability 1.*

*Proof of Lemma 13.* To begin, observe that:

$$\sup_{m(t) < h \leq m(t+T)} \| \sum_{i=m(t)}^{h-1} \gamma_{i+1} (U_{i+1} + b_{i+1})\| \leq \sup_{m(t) < h \leq m(t+T)} \| \sum_{i=m(t)}^{h-1} \gamma_{i+1} U_{i+1}\|$$

$$+ \sup_{m(t) < h \leq m(t+T)} \| \sum_{i=m(t)}^{h-1} \gamma_{i+1} b_{i+1}\|$$

$$\leq \sup_{m(t) < h \leq m(t+T)} \| \sum_{i=m(t)}^{h-1} \gamma_{i+1} U_{i+1}\|$$

$$+ (T+1) \max_{m(t) < h \leq m(t+T)} b_{h+1}$$

By definition, $T \max_{m(t) < h \leq m(t+T)} b_{h+1}$ goes to 0. Thus, it remains to prove that $\lim_{t \to \infty} \sup_{m(t) < h \leq m(t+T)} \|\sum_{i=m(t)}^{h-1} \gamma_{i+1} U_{i+1}\| = 0$ with probability 1.

For any $t \geq 0$ Burkholder's inequality implies that, for some universal constant $C_q > 0$:

$$\mathbb{E}\left[ \sup_{m(kT) < h \leq m((k+1)T)} \| \sum_{i=m(kT)}^{h-1} \gamma_{i+1} U_{i+1}\|^q \right] \leq C_q \mathbb{E}\left[ \left( \sum_{i=m(kT)}^{m((k+1)T)-1} \gamma_{i+1}^2 \|U_{i+1}\|^2 \right)^{\frac{q}{2}} \right] \quad \text{(B.2)}$$

If $q = 2$ we directly obtain:

$$\mathbb{E}\left[ \sup_{m(kT) < h \leq m((k+1)T)} \| \sum_{i=m(kT)}^{h-1} \gamma_{i+1} U_{i+1}\|^2 \right] \leq C_2 \sum_{i=m(kT)}^{m((k+1)T)-1} \gamma_{i+1}^2 \mathbb{E}\left[\|U_{i+1}\|^2\right]$$

For $q > 2$ we need the Hölder's inequality :

$$\sum_i x_i y_i \leq \left( \sum_i x_i^u \right)^{\frac{1}{u}} \left( \sum_i y_i^{\frac{u}{u-1}} \right)^{\frac{u-1}{u}}$$

which applied to $x_i = \alpha_i^{1-\delta}|\beta_i|$ and $y_i = \alpha_i^\delta$ gives for any $\alpha_i \geq 0$, $\beta_i \in \mathbb{R}$, $u > 1$ and $0 < \delta < 1$:

$$\left( \sum_i |\alpha_i \beta_i| \right)^u \leq \left( \sum_i \alpha_i^{\frac{\delta u}{u-1}} \right)^{u-1} \sum_i \alpha_i^{(1-\delta)u} |\beta_i|^u \quad \text{(B.3)}$$

Suppose $q > 2$ we apply B.3 to the second member of B.2 with $u = \frac{q}{2}$, $\delta = \frac{q-2}{2q}$, $\alpha_i = \gamma_{i+1}^2$ and $\beta_i = \|U_{i+1}\|^2$:

$$\mathbb{E}\left[ \sup_{m(kT) < h \leq m((k+1)T)} \| \sum_{i=m(kT)}^{h-1} \gamma_{i+1} U_{i+1}\|^q \right] \leq C_q \mathbb{E}\left[ \left( \sum_{i=m(kT)}^{m((k+1)T)-1} \gamma_{i+1} \right)^{\frac{q}{2}-1} \sum_{i=m(kT)}^{m((k+1)T)-1} \gamma_{i+1}^{\frac{q}{2}+1} \|U_{i+1}\|^q \right]$$

$$\leq C_q T^{\frac{q}{2}-1} \sum_{i=m(kT)}^{m((k+1)T)-1} \gamma_{i+1}^{\frac{q}{2}+1} \mathbb{E}[\|U_{i+1}\|^q]$$

$$\text{(B.4)}$$

Therefore, for $q \geq 2$, we have

$$\mathbb{E}\left[\sup_{m(kT)<h\leq m((k+1)T)} \|\sum_{i=m(kT)}^{h-1} \gamma_{i+1}U_{i+1}\|^q\right] \leq C_q T^{\frac{q}{2}-1} \sum_{i=m(kT)}^{m((k+1)T)-1} \gamma_{i+1}^{\frac{q}{2}+1} \mathbb{E}[\|U_{i+1}\|^q]. \quad (B.5)$$

Let $E(k,\varepsilon)$ be the event $A(kt,T) = \sup_{m(kt)<h\leq m(kt+T)}\|\sum_{i=m(t)}^{h-1} \gamma_{i+1}U_{i+1}\| > \varepsilon$, for $\varepsilon > 0$.
Doob's martingal inequality yiels:

$$\mathbb{P}(E(k,\varepsilon)) \leq \frac{\mathbb{E}[A(kT,T)^q]}{\varepsilon^q}$$

So

$$\begin{aligned}\sum_{k\in\mathbb{N}} \mathbb{P}(E(k,\varepsilon)) &\leq \frac{\sum_{k\geq 0}\mathbb{E}[A(kT,T)^q]}{\varepsilon^q} \\ &= \frac{C_q T^{\frac{q}{2}-1}\sum_{i=0}^{\infty}\gamma_{i+1}^{(q/2+1)}\mathbb{E}[\|\overline{U}_{i+1}\|^q]}{\varepsilon^q} \\ &< \infty \end{aligned} \quad (B.6)$$

By the Borel-Cantelli Lemma this proves that for all $\varepsilon > 0$:

$$\mathbb{P}(\limsup_{k\to\infty} E(k,\varepsilon)) = 0$$

with probability one. So for all $\varepsilon$ the set of $k$ such that so $E(k,\varepsilon)$ is true is a null set.

Consider now $E(n) = \{k : E(k,\frac{1}{n}) \text{ is true}\}$. For all $n$, $E(n)$ is a null set. Therefore as a countable union of null set $\bigcup_{n\in\mathbb{N}} E(n)$ is a null set, so $\lim_{k\to\infty} A(kT,T) = 0$ with probability one.

On the other hand for $kT \leq t < (k+1)T$:

$$\begin{aligned}A(t,T) &= \sup_{m(t)<h\leq m(t+T)}\|\sum_{i=m(t)}^{h-1} \gamma_{i+1}U_{i+1}\|^q \\ &\leq \max\left(\sup_{m(t)<h\leq m((k+1)T)}\|\sum_{i=m(t)}^{h-1} \gamma_{i+1}U_{i+1}\|^q, \sup_{m((k+1)T)<h\leq m(t+T)}\|\sum_{i=m(t)}^{h-1} \gamma_{i+1}U_{i+1}\|^q\right) \\ &= \max\left(\sup_{m(t)<h\leq m((k+1)T)}\|\sum_{i=m(kT)}^{h-1}\gamma_{i+1}U_{i+1} - \sum_{i=m(kT)}^{m(t)-1}\gamma_{i+1}U_{i+1}\|^q,\right. \\ &\quad \left.\sup_{m((k+1)T)<h\leq m(t+T)}\|\sum_{i=m(kT)}^{m((k+1)T)-1}\gamma_{i+1}U_{i+1} - \sum_{i=m(kT)}^{m(t)-1}\gamma_{i+1}U_{i+1} + \sum_{i=m((k+1)T)}^{h-1}\gamma_{i+1}U_{i+1}\|^q\right) \\ &\leq 2A(kT,T) + A((k+1)T,T)\end{aligned} \quad (B.7)$$

This concludes the proof. $\qquad\square$

Now, let $L = \max_{x\in\mathcal{X},i\in\mathcal{N}} u_i(x)$, and $A_i = |\mathcal{A}_i|$.

$$\hat{v}_{i\alpha_i}(n) = \frac{\mathbb{1}_{\alpha_i=\alpha_i(n)}}{X_{i\alpha_i}(n-1)} u_i(\alpha(n)) \quad (B.8)$$

First it is useful to remark that we have :

$$\mathbb{E}[\hat{v}_{i\alpha_i}(n)\,|\,\mathcal{F}_{n-1}] = \mathbb{E}\left[\frac{\mathbb{1}_{\alpha_i=\alpha_i(n)}}{X_{i\alpha_i}(n-1)} u_i(\alpha(n))\,\bigg|\,\mathcal{F}_{n-1}\right] = v_{i\alpha_i}(X(n-1))$$

$$\mathbb{E}\left[\left(\hat{v}_{i\alpha_i}(n) - v_{i\alpha_i}(X(n-1))\right)^2 \middle| \mathcal{F}_{n-1}\right] \le 2\,\mathbb{E}\left[\frac{\mathbb{1}_{\alpha_i=\alpha_i(n)}}{X_{i\alpha_i}(n-1)^2} u_i^2(\alpha(n)) \middle| \mathcal{F}_{n-1}\right] + 2\,\mathbb{E}\left[v_{i\alpha_i}^2(X(n-1)) \middle| \mathcal{F}_{n-1}\right]$$

$$\le 2\frac{L^2 A_i}{\varepsilon_{n-1}} + 2L^2 = 2L^2\left(1 + \frac{A_i}{\varepsilon_{n-1}}\right)$$

$$\text{(B.9)}$$

We begin by proving that when a strategies always leads to a better payoff, than an other its score becomes higher. So omitting the player index we can prove the following Lemma.

**Lemma 14.** *Suppose that Algorithm 1 is run with i) a bandit feedback estimator B.8; ii) a decreasing step-size $\gamma_n$; iii) a decreasing mixing factor $\varepsilon_n$ such that :*

$$\lim_{n\to\infty} \gamma_n = \infty \text{ and } \sum_{n=1}^{\infty} \frac{1}{\varepsilon_n n^2} < \infty \qquad \text{(B.10)}$$

*If there exists some $\mu > 0$ and a set $V \subset \mathcal{X}$ such that $v_\beta(x) - v_\alpha(x) \ge \mu$ for all $x \in V$ and if $X(n) \in V$, for all $n$, then for all $c \in (0, \mu)$, there exists some $T_0$ such that $Y_\beta(T) - Y_\alpha(T) \ge c\sum_{n=1}^{T} \gamma_n$ for all $T \ge T_0$ (a.s.).*

*Proof.* Proof of Lemma 14 Let $\zeta_k = \hat{v}_\beta(k) - v_\beta(X(k-1)) - [\hat{v}_\alpha(k) - v_\alpha(X(k-1))]$. By assumption there exists $a > 0$ such that $v_\beta(x) - v_\alpha(x) \ge a$ for all $x \in V$. Then,

$$Y_\beta(T) - Y_\alpha(T) = c_{\beta\alpha} + \sum_{n=1}^{T} \gamma_n \left[\hat{v}_\beta(n) - \hat{v}_\alpha(X(n-1))\right]$$

$$= c_{\beta\alpha} + \sum_{n=1}^{T} \gamma_n \left[v_\beta(X(n-1)) - v_\alpha(X(n-1))\right] + \sum_{n=1}^{T} \gamma_n \zeta_n \qquad \text{(B.11)}$$

$$\ge c_{\beta\alpha} + \tau_T \left[\mu + \frac{\sum_{n=1}^{T} \gamma_n \zeta_n}{\tau_T}\right] \qquad \text{(B.12)}$$

where we set $c_{\beta\alpha} = Y_\beta(0) - Y_\alpha(0)$ and $\tau_T = \sum_{n=1}^{T} \gamma_n$.

The law of large numbers for martingale difference sequences [31, Theorem 2.18] gives $\tau_T^{-1} \sum_{n=1}^{T} \gamma_n \zeta_n \to 0$ (a.s.), provided that $\sum_{n=1}^{\infty} \frac{\mathbb{E}[\|\gamma_n \zeta_n\|^2 | \mathcal{F}_{n-1}]}{\tau_n^2} < \infty$ and $\sum_{n=1}^{\infty} \gamma_n = \infty$.

We focus on an upper bound to $\mathbb{E}\left[\|\gamma_n \zeta_n\|^2 \middle| \mathcal{F}_{n-1}\right]$, using (B.9) we obtain:

$$\mathbb{E}\left[\|\gamma_n \zeta_n\|^2 \middle| \mathcal{F}_{n-1}\right] \le 4\gamma_n^2 \max_{\alpha \in \mathcal{A}} \mathbb{E}\left[(\hat{v}_\alpha(n) - v_\alpha)^2 \middle| \mathcal{F}_{n-1}\right]$$

$$\le 8L^2 \gamma_n^2 \left(\frac{A}{\varepsilon_{n-1}} + 1\right)$$

$$\text{(B.13)}$$

Using that $\tau_n^2 \ge n^2 \gamma_n^2$ we obtain:

$$\frac{\mathbb{E}\left[\|\gamma_n \zeta_n\|^2 \middle| \mathcal{F}_{n-1}\right]}{\tau_n^2} \le \frac{8L^2}{n^2}\left(1 + \frac{A}{\varepsilon_{n-1}}\right)$$

and

$$\sum_{n=1}^{T} \frac{\mathbb{E}\left[\|\gamma_n \zeta_n\|^2 \middle| \mathcal{F}_{n-1}\right]}{\tau_n^2} \le 8L^2 \left(\sum_{n=1}^{T} \frac{1}{n^2} + \sum_{n=1}^{T} \frac{A}{\varepsilon_{n-1} n^2}\right) = \mathcal{O}(\sum_{n=1}^{T} \frac{1}{\varepsilon_{n-1} n^2}) < \infty$$

We readily get $\tau_T^{-1} \sum_{n=1}^{T} \gamma_n \zeta_n \to 0$ with probability 1. As a result, for all $c \in (0, \mu)$, there exists some random (but a.s. finite) $T_0$ such that if $T \ge T_0$, then $Y_\beta(T) - Y_\alpha(T) \ge c\tau_T$. $\qquad \square$

With this auxiliary result at hand, our proof of Theorem 4 relies on the following four steps:

1. Show that $X$ is an asymptotic pseudo trajectory of a continuous dynamics (Proposition 17),

2. Show that the potential function of the game is a strict Lyapunov function of the dynamics (Proposition 18),

3. Show that $X$ converges towards a rest point of the dynamics (Proposition 19),

4. Show that if $X$ converged toward a point it is a Nash Equilibrium (Proposition 20).

**Definition 15** (Asymptotic Pseudo-trajectories). Given a flow $\phi : \mathbb{R} \times M \to M, (n, x) \to \phi(n, x) = \phi_n(x)$ such that $\phi_0 = $ Identity and $\phi_{n+\alpha} = \phi_n \circ \phi_\alpha$, a continuous function $X : \mathbb{R} \to M$ is an asymptotic pseudo-trajectory if

$$lim_{n\to\infty} \sup_{0 \le k \le T} d((X(n+k), \phi_h(X(n))) = 0 \text{ for any } T > 0 \tag{B.14}$$

Before addressing the proof we show that specific conditions on the mixing factor and the step size, guarantee a few important relations.

**Lemma 16.** *Suppose that Algorithm 1 is run with i) the bandit estimator* (B.8)*, ii) a step size sequence $\gamma_n$ decreasing to 0 and iii) a strictly positive mixing factor $\varepsilon_n$ decreasing to 0 such that:*

$$\lim_{n\to\infty} \frac{\gamma_n}{\varepsilon_n^2} = 0, \ \sum_{n=1}^{\infty} \frac{\gamma_n^2}{\varepsilon_n} < \infty \tag{B.15}$$

*Then for all players $i$ and $\alpha \in \mathcal{A}_i$, $U_{i\alpha}(n+1) = \nabla \Lambda_{i\alpha}^{\mathrm{T}}(Y_i(n)) (\hat{v}_i(n+1) - v_i(X(n)))$, is a martingale difference noise, $\sum_n \gamma_{n+1}^2 \mathbb{E}[\|U_{i\alpha}(n+1)\|^2] < \infty$ and $\lim_{n\to\infty} \gamma_{n+1}\hat{v}_{i\alpha}(n+1)^2 = 0$.*

*Proof.* Proof of Lemma 16 For all players $i \in \mathcal{N}$, and for all $\alpha \in \mathcal{A}_i$, $U_{i\alpha}(n)$ is a martingale difference noise. In deed, $\mathbb{E}[U_{i\alpha}(n+1) | \mathcal{F}_n] = \nabla \Lambda_{i\alpha}^{\mathrm{T}}(Y_i(n)) (v_i(X(n)) - v_i(X(n))) = 0$ (because $\mathbb{E}[\hat{v}_i(n+1) | \mathcal{F}_n] = v_i(X(n))$), and by definition of the bandit estimator, $\mathbb{E}\left[(\hat{v}_{i\alpha_i}(n+1) - v_{i\alpha_i}(X(n)))^2 \,\middle|\, \mathcal{F}_n\right]$ is finite for all $n$, so $\mathbb{E}[\|U_{i\alpha}(n+1)\|^2]$ is also finite for all $n$, so $U_{i\alpha}(n)$ is a martingale difference noise.

To prove that $\sum_n \gamma_{n+1}^2 \mathbb{E}[\|U_{i\alpha}(n+1)\|^2] < \infty$, remark that $\mathbb{E}[\|U_{i\alpha}(n+1)\|^2 \,|\, \mathcal{F}_n] = \mathcal{O}(\frac{1}{\varepsilon_n})$. Therefore $\sum_{n=1}^{\infty} \frac{\gamma_n^2}{\varepsilon_{n-1}} < \infty$ allows us to conclude the second point.

For the last point, observe that $\hat{v}_{i\alpha}(n+1) = \mathcal{O}(\frac{1}{\varepsilon_n^2})$, so the hypothesis $\lim_{n\to\infty} \frac{\gamma_n}{\varepsilon_{n-1}^2} = 0$ is sufficient to conclude that $\lim_{n\to\infty} \gamma_{n+1}\hat{v}_{i\alpha}(n+1)^2 = 0$. $\qquad \square$

**Proposition 17.** *Let $\Gamma$ be a generic potential game and suppose that Algorithm 1 is run with i) the bandit estimator* (B.8)*, ii) a step size sequence $\gamma_n$ decreasing to 0 and iii) a strictly positive mixing factor $\varepsilon_n$ decreasing to 0 such that:*

$$\sum_{n=1}^{\infty} \gamma_n = \infty, \ \lim_{n\to\infty} \frac{\gamma_n}{\varepsilon_n^2} = 0, \ \sum_{n=1}^{\infty} \frac{\gamma_n^2}{\varepsilon_n} < \infty \text{ and } \lim_{n\to\infty} \frac{\varepsilon_n - \varepsilon_{n+1}}{\gamma_{n+1}} = 0. \tag{B.16}$$

*Then, for all players $i \in \mathcal{N}$, the interpolated process of the sequences $(X_i(n))_{n\in\mathbb{N}}$ are asymptotic pseudo trajectories of replicator dynamics:*

$$\dot{x}_{i\alpha} = x_{i\alpha}\left[ v_{i\alpha}(x) - \sum_{\beta\in\mathcal{A}_i} x_{i\beta}v_{i\beta}(x) \right], \tag{RD}$$

*Proof.* Proof of Proposition 17 Observe that for any $i \in \mathcal{N}$, for any $\alpha, \beta, \beta' \in \mathcal{A}_i$, $\Lambda_{i\alpha}(y_i) = \frac{\exp(y_{i\alpha})}{\sum_{s\in\mathcal{A}_i} \exp(y_{i\alpha})}$, we have

$\frac{\partial \Lambda_{i\alpha}(y_i)}{\partial y_{i\beta}} = \Lambda_{i\alpha}(y_i)(\mathbb{1}_{\alpha=\beta} - \Lambda_{i\beta}(y_i))$, and $\frac{\partial^2 \Lambda_{i\alpha}(y_i)}{\partial y_{i\beta}\partial y_{i\beta'}} = \Lambda_{i\alpha}(y_i) (\mathbb{1}_{\alpha=\beta=\beta'} - \mathbb{1}_{\alpha=\beta} \Lambda_{i\beta'}(y_i) - \Lambda_{i\beta}(y_i)(\mathbb{1}_{\alpha=\beta'} + \mathbb{1}_{\beta=\beta'} - 2\Lambda_{i\beta'}(y_i)))$.

Using Taylor's Remainder Theorem, we rewrite the equation $X_{i\alpha}(n+1) = \frac{\varepsilon_{n+1}}{A_i} + (1 - \varepsilon_{n+1}) \Lambda_{i\alpha}(Y_i(n+1))$ as

$$
\begin{aligned}
X_{i\alpha}(n+1) ={}& \frac{\varepsilon_{n+1}}{A_i} + (1 - \varepsilon_{n+1}) \Lambda_{i\alpha}(Y_i(n) + \gamma_{n+1}\hat{v}_i(n+1)) \\
={}& \frac{\varepsilon_{n+1}}{A_i} + (1 - \varepsilon_{n+1}) \Lambda_{i\alpha}(Y_i(n)) \\
& + (1 - \varepsilon_{n+1})\gamma_{n+1} \left( \nabla \Lambda_{i\alpha}^{\mathrm{T}}(Y_i(n))\hat{v}_i(n+1) + \frac{1}{2}\gamma_{n+1}\hat{v}_i^{\mathrm{T}}(n+1) \operatorname{Hess} \Lambda_{i\alpha}(\psi_i(n))\hat{v}_i(n+1) \right) \\
={}& X_{i\alpha}(n) + (\varepsilon_{n+1} - \varepsilon_n) \left( \frac{1}{A_i} - \Lambda_{i\alpha}(Y_i(n)) \right) \\
& + (1 - \varepsilon_{n+1})\gamma_{n+1} \left( \nabla \Lambda_{i\alpha}^{\mathrm{T}}(Y_i(n))\hat{v}_i(n+1) + \frac{\gamma_{n+1}}{2}\hat{v}_i^{\mathrm{T}}(n+1) \operatorname{Hess} \Lambda_{i\alpha}(\psi_i(n))\hat{v}_i(n+1) \right) \\
={}& X_{i\alpha}(n) + (1 - \varepsilon_{n+1})\gamma_{n+1} \left( \nabla \Lambda_{i\alpha}^{\mathrm{T}}(Y_i(n))\hat{v}_i(n+1) + a_{in} \right) \\
={}& X_{i\alpha}(n) + (1 - \varepsilon_{n+1})\gamma_{n+1} \left( \nabla \Lambda_{i\alpha}^{\mathrm{T}}(Y_i(n))v_i(X(n)) + \nabla \Lambda_{i\alpha}^{\mathrm{T}}(Y_i(n)) \left( \hat{v}_i(n+1) - v_i(X(n)) \right) + a_{in} \right)
\end{aligned}
$$
(B.17)

where $\nabla \Lambda_{i\alpha}$ is the gradient vector of $\Lambda_{i\alpha}$, $\nabla \Lambda_{i\alpha}^{\mathrm{T}}$ is its transposed, $\operatorname{Hess} \Lambda_{i\alpha}$ is the Hessian matrix of $\Lambda_{i\alpha}$, and $\psi_i(n)$ is in the line segment going out from $Y_i(n)$ to the point $Y_i(n+1)$ and $a_{in} = \frac{\gamma_{n+1}}{2} \hat{v}_i^{\mathrm{T}}(n+1) \operatorname{Hess} \Lambda_{i\alpha}(\psi_i(n))\hat{v}_i(n+1) + \frac{\varepsilon_{n+1} - \varepsilon_n}{(1 - \varepsilon_{n+1})\gamma_{n+1}} \left( \frac{1}{A_i} - \Lambda_{i\alpha}(Y_i(n)) \right)$.

Furthermore, $\nabla \Lambda_{i\alpha}^{\mathrm{T}}(Y_i(n))v_i(X(n)) = \Lambda_{i\alpha}(Y_i(n)) \left( v_{i\alpha}(X(n)) - \sum_{\beta \in \mathcal{A}_i} \Lambda_{i\beta}(Y_i(n))v_{i\beta}(X(n)) \right)$ and we can rewrite

$$
\Lambda_{i\alpha}(Y_i(n)) = \frac{X_{i\alpha}(n) - \frac{\varepsilon_n}{A_i}}{1 - \varepsilon_n} = X_{i\alpha}(n) + \frac{\varepsilon_n}{1 - \varepsilon_n} \left( X_{i\alpha}(n) - \frac{1}{A_i} \right)
$$

Therefore,

$$
\begin{aligned}
\nabla \Lambda_{i\alpha}^{\mathrm{T}}(Y_i(n))v_i(X(n)) ={}& X_{i\alpha} \left( v_{i\alpha}(X(n)) - \sum_{\beta \in \mathcal{A}_i} X_{i\beta}(n)v_{i\beta}(X(n)) \right) \\
& + \frac{\varepsilon_n}{1 - \varepsilon_n} \left[ \frac{\sum_{\beta \in \mathcal{A}_i}(X_{i\alpha}(n) + X_{i\beta}(n))v_{i\beta}(X(n))}{A_i} \right] \\
& - 2\frac{\varepsilon_n}{1 - \varepsilon_n} \left[ \sum_{\beta \in \mathcal{A}_i} X_{i\alpha}(n)X_{i\beta}(n)v_{i\beta}(X(n)) \right] \\
& + \frac{\varepsilon_n}{1 - \varepsilon_n} \left[ v_{i\alpha}(X(n))X_{i\alpha}(n) - \frac{v_{i\alpha}(X(n))}{A_i} \right] \\
& - \frac{\varepsilon_n^2}{(1 - \varepsilon_n)^2} \sum_{\beta \in \mathcal{A}_i} \left( X_{i\alpha}(n) - \frac{1}{A_i} \right) \left( X_{i\beta}(n) - \frac{1}{A_i} \right) v_{i\beta}(X(n))
\end{aligned}
$$
(B.18)

Next, we focus on $b_{in} = a_{in} + \nabla \Lambda_{i\alpha}^{\mathrm{T}}(Y_i(n))v_i(X(n)) - X_{i\alpha} \left( v_{i\alpha}(X(n)) - \sum_{\beta \in \mathcal{A}_i} X_{i\beta}(n)v_{i\beta}(X(n)) \right)$ and show that it converges towards $0$ as $n$ grows. The term $\nabla \Lambda_{i\alpha}^{\mathrm{T}}(Y_i(n))v_i(X(n)) = X_{i\alpha} \left( v_{i\alpha}(X(n)) - \sum_{\beta \in \mathcal{A}_i} X_{i\beta}(n)v_{i\beta}(X(n)) \right)$ goes to $0$ because $\varepsilon_n$ is decreasing. Since $\frac{\partial^2 \Lambda_{i\alpha}(y_i)}{\partial y_{i\beta} \partial y_{i\beta'}} = \Lambda_{i\alpha}(y_i)(\mathbb{1}_{\alpha=\beta=\beta'} - \mathbb{1}_{\alpha=\beta} \Lambda_{i\beta'}(y_i) - \Lambda_{i\beta}(y_i)(\mathbb{1}_{\alpha=\beta'} + \mathbb{1}_{\beta=\beta'} - 2\Lambda_{i\beta'}(y_i)))$, all components of $\operatorname{Hess} \Lambda_{i\alpha}(\psi_i(n))$ are bounded. In addition $\lim_{n \to \infty} \frac{\varepsilon_n - \varepsilon_{n+1}}{\gamma_{n+1}} = 0$. So, the limit of $b_{in}$ (when $n \to \infty$) depends on the limit of $\lim_{n \to \infty} \gamma_n \|\hat{v}_{i\alpha}(n)\|^2 = 0$ as proved in Lemma 16.

We can write the dynamics of Algorithm 1 in the form:

$$
X_{i\alpha}(n+1) = X_{i\alpha}(n) + \gamma_{n+1} \left( F_{i\alpha}(n) + U_{i\alpha}(n+1) + b_{in} \right)
$$

where $F_{i\alpha}(n) = X_{i\alpha}\left(v_{i\alpha}(X(n)) - \sum_{\beta \in \mathcal{A}_i} X_{i\beta}(n)v_{i\beta}(X(n))\right)$ is the replicator dynamics, and $U_{i\alpha}(n+1) = \nabla \Lambda_{i\alpha}^{\mathrm{T}}(Y_i(n))\,(\hat{v}_i(n+1) - v_i(X(n)))$.

In Lemma 16 we also showed that $U_{i\alpha}(n)$ is a martingale difference noise and that $\sum_n \gamma_{n+1}^2 \mathbb{E}\left[\|U_{n+1}\|^2\right] < \infty$. We apply Lemma 13, and Proposition 4.1 of [28] allows us to conclude that the interpolated process of the sequences $(X_i(n))_{n\in\mathbb{N}}$ are asymptotic pseudo trajectories of the replicator dynamics.

$\square$

**Proposition 18.** *Let $\Gamma$ be a generic potential game. The potential function $f$ of $\Gamma$ is a strict increasing Lyapunov function of the flow inducted by the replicator dynamics.*

*Proof.* Proof of Proposition 18 We consider the variation of $f$ to show that $f$ is an increasing function. Using the definition of potential function, we have:

$$
\begin{aligned}
\dot{f}(x) &= \sum_{i\in\mathcal{N}} \sum_{\alpha\in\mathcal{A}_i} \frac{\partial f}{\partial x_{i\alpha}}(x)\dot{x}_{i\alpha} \\
&= \sum_{i\in\mathcal{N}} v_i^{\mathrm{T}}(x(t))\dot{x}_i(t) \\
&= \sum_{i\in\mathcal{N}} \sum_{\alpha\in\mathcal{A}_i} v_{i\alpha}(x(t))\dot{x}_{i\alpha}(t) \\
&= \sum_{i\in\mathcal{N}} \sum_{\alpha\in\mathcal{A}_i} v_{i\alpha}(x(t))x_{i\alpha}(t)\left(v_{i\alpha}(x(t)) - \sum_{\beta\in\mathcal{A}_i} v_{i\beta}(x(t))x_{i\beta}(t)\right) \\
&= \sum_{i\in\mathcal{N}} \sum_{\alpha\in\mathcal{A}_i} \sum_{\beta\in\mathcal{A}_i,\beta>\alpha} x_{i\alpha}(t)x_{i\beta}(t)[v_{i\alpha}(x(t)) - v_{i\beta}(x(t))]^2 \\
&\geq 0
\end{aligned}
\tag{B.19}
$$

Furthermore, when $x$ is a rest point of the dynamics, $\dot{x} = 0$ and $\dot{f}(x) = 0$. Conversely, we prove that $\dot{f}(x) = 0$ implies that $x$ is a rest point of the dynamics.

Observe that $\dot{f}(x) = 0$ then for all players $i$ in $\mathcal{N}$, and for all pure strategies $\alpha$ and $\beta$ in $\mathcal{A}_i$, $x_{i\alpha}(t) = 0$ or $x_{i\beta}(t) = 0$ or $v_{i\alpha}(x(t)) = v_{i\beta}(x(t))$. This also leads to $\dot{x}_{i\alpha}$ for all players $i$ in $\mathcal{N}$, and for all pure strategies $\alpha$ in $\mathcal{A}_i$. Therefore $x$ is a rest point of the dynamics, and $f$ is a strict increasing Lyapunov function.

$\square$

**Proposition 19.** *Let $\Gamma$ be a generic potential game and suppose that Algorithm 1 is run with i) the bandit estimator (B.8), ii) a step size sequence $\gamma_n$ decreasing to $0$ and iii) a strictly positive mixing factor $\varepsilon_n$ decreasing to $0$ such that:*

$$
\sum_{n=1}^{\infty} \gamma_n = \infty\,,\quad \lim_{n\to\infty} \frac{\gamma_n}{\varepsilon_n^2} = 0\,,\quad \sum_{n=1}^{\infty} \frac{\gamma_n^2}{\varepsilon_n} < \infty\ \text{and}\ \lim_{n\to\infty} \frac{\varepsilon_n - \varepsilon_{n+1}}{\gamma_{n+1}} = 0.
\tag{B.20}
$$

*Then interpolated process of the sequences $X(n)$ converges (a.s.) to a rest point of the replicator dynamics.*

*Proof.* Proof of Proposition 19 We showed in Proposition 17 that under the same assumptions interpolated process of the sequences $X(n)$ is a pseudo asymptotic trajectory of the flow induced by the replicator dynamics.

We now show that the replicator dynamics has a finite number of rest points. The dynamics is

$$
\dot{x}_{i\alpha} = x_{i\alpha}\left[v_{i\alpha}(x) - \sum_{\beta\in\mathcal{A}_i} x_{i\beta}v_{i\beta}(x)\right]
$$

So $x$ is a rest point if and only if $v_{i\beta}(x) = v_{i\alpha}(x), \forall \alpha, \beta \in \mathrm{supp}(x_i)$. Therefore $x$ is a rest point of the replicator dynamics if and only if it is a restricted Nash-equilibrium. The game is finite so they are a finite number of game restrictions. Each restricted game is a finite, generic and potential game so it has a finite number of Nash-equilibrium. Therefore the number of rest points is finite.

Corollary 6.6 of [28] allows to conclude that the linear interpolation of the sequences $X(n)$ converges (a.s.) to a rest point of the replicator dynamics.

$\square$

**Proposition 20.** *Suppose that Algorithm 1 is run with i) the bandit estimator (B.8), ii) a step size sequence $\gamma_n$ decreasing to $0$ and iii) a strictly positive mixing factor $\varepsilon_n$ decreasing to $0$ such that:*

$$\sum_{n=1}^{\infty} \gamma_n = \infty \,, \quad \lim_{n\to\infty} \frac{\gamma_n}{\varepsilon_n^2} = 0 \,, \quad \sum_{n=1}^{\infty} \frac{1}{\varepsilon_n n^2} < \infty \,, \quad \sum_{n=1}^{\infty} \frac{\gamma_n^2}{\varepsilon_n} < \infty \ and \ \lim_{n\to\infty} \frac{\varepsilon_n - \varepsilon_{n+1}}{\gamma_{n+1}} = 0. \quad \text{(B.21)}$$

*Then if $X(n)$ converges, its limit is a Nash Equilibrium of the game.*

*Proof.* Proof of Proposition 20 Let $x^* \in X$ such that $\lim_{n\to\infty} X(n) = x^*$, we show by contradiction that $x^*$ is a Nash Equilibrium. Suppose the contrary, by definition of a Nash Equilibrium we have :

$$\exists i \in \mathcal{N}, \exists \beta \in \mathcal{A}_i, \beta \notin \mathrm{supp}(x_i^*), s.t., u_{i\beta}(x^*) > u_{i\alpha}(x^*), \forall \alpha \in \mathrm{supp}(x_i^*)$$

By continuity of the utility function $u$, there exists a neighborhood $U$ of $x^*$ and $a > 0$ such that:

$$\exists i \in \mathcal{N}, \exists \beta \in \mathcal{A}_i, \beta \notin \mathrm{supp}(x_i^*), s.t., u_{i\beta}(x) - u_{i\alpha}(x) > a, \forall \alpha \in \mathrm{supp}(x_i^*), \forall x \in U$$

For $n$ big enough, $X(n) \in U$, so using Lemma 14, for $n_0$ big enough and for $n \geq n_0$ we have :

$$Y_{i\beta}(T) - Y_{i\alpha}(T) \geq c \sum_{n=1}^{T} \gamma_n \quad \text{for all } T \geq T_0 \text{ (a.s.)}.$$

Thus, with $X_{i\alpha}(T) = \frac{\varepsilon(T)}{A_i} + (1 - \varepsilon(T)) \Lambda_{i\alpha}(Y_i(T))$ by definition, we get

$$\begin{aligned}
X_{\alpha}(T) &= \frac{\varepsilon(T)}{A_i} + (1 - \varepsilon(T)) \frac{e^{Y_{i\alpha}(T)}}{\sum_{\beta'} e^{Y_{i\beta'}(T)}} \\
&\leq \frac{\varepsilon(T)}{A_i} + (1 - \varepsilon(T)) \frac{e^{Y_{i\alpha}(T)}}{e^{Y_{i\beta}(T)}} \\
&= \frac{\varepsilon(T)}{A_i} + (1 - \varepsilon(T)) e^{Y_{i\alpha}(T) - Y_{i\beta}(T)} \\
&\leq \frac{\varepsilon(T)}{A_i} + (1 - \varepsilon(T)) e^{-c \sum_{n=1}^{T} \gamma_n} \text{(a.s.)},
\end{aligned} \qquad \text{(B.22)}$$

So $\lim_{T\to\infty} X_{i\alpha}(T) = 0$ which is a contradiction with $\alpha \in \mathrm{supp}(x_i^*)$. $\square$