[Reviews · NeurIPS 2017]

Reviewer 1



SUMMARY The paper scrutinizes the concept of a Nash equilibrium. In non-cooperative games it is often difficult to compute Nash equilibria. Furthermore, it might be unreasonable to assume full rationality and many other properties that are commonly assumed. So is Nash equilibrium a good concept? Specifically, the authors ask: if all players of a repeated game follow a no-regret learning algorithm, does play converge to a Nash equilibrium? The authors focus on variants of the exponential weights algorithm in potential games. In this context they consider two different feedback models: semi-bandit and bandit. In the semi-bandit case the authors show, under mild conditions, that learning converges to a Nash equilibrium at a quasi-exponential rate. In the bandit case similar results hold when the algorithm is run with a positive mixing factor, to obtain convergence to an epsilon-Nash equilibrium. By letting the mixing factor epsilon go to zero at a suitable rate, it is possible to obtain convergence to an actual Nash equilibrium. The proofs are based on stochastic approximation techniques. REVIEW The paper is very well written, it was a pleasure to read. The problem is well motivated, relevant, and interesting. The results of the paper are of significant interest to the NIPS community. I am confident that the results are correct --- the high level proof sketches in the main text are certainly convincing. I do not have any feedback to the authors regarding the main text. However, I recommend that the authors take a few careful passes on the supplementary material, which could be written better.

Reviewer 2



The authors show that exponential weight updates and its generalization FTRL, converges pointwise to a Nash equilibrium in potential games. The latter holds even when players receive bandit feedback of the game. Unfortunately, the main result, for the case of full feedback is already known in the economics literature, where Hedge is known as smooth fictitious play. It is presented in the paper: Hofbauer, Sandholm, "On the global convergence of smooth fictitious play", Econometrica 2002. This work is not even cited in the paper and it shows that smooth fictitious play and its extensions of FTRL (though they do not call it FTRL) does converge to Nash in potential games and in three other classes of games. Moreover, the result of Kleinberg, Piliouras, Tardos:"Multiplicative Updates Outperform Generic No-Regret Learning in Congestion Games", STOC'09, is presented in the paper, for the actual sequence of play (unlike what the authors claim). So that paper shows that for the class of congestion games (which is equivalent to the class of all potential games (c.f. Monderer-Shapley)), the actual play of MWU converges to Nash equilibria and in fact almost surely to Pure Nash equilibria (except for measure zero game instances), which is an even stronger result. In light of the above two omissions, the paper needs to be re-positioned to argue that the extension to bandit feedback is not an easy extension of the above results. It is an interesting extension and I can potentially see that extension be presented at a NIPS quality conference, but it needs to be positioned correctly with the literature and the paper needs to argue as to why it does not easily follow from existing results. In particular, given that most of these results go through stochastic approximations, such stochastic approximations are most probably robust to unbiased estimates of the payoffs, which is all that is needed for the bandit extension.

Reviewer 3



The paper investigates the exponential weighting algorithm in no-regret situation, a model free approach, for arriving at Nash equilibrium, as opposed to previous investigations which guaranteed with vanishing regret based on definition of egret. No-regret dynamics exhibit good convergence if all players adopt this strategy. The paper considers both semi-bandit and bandit cases. It proved that the EW scheme converges to relaxed Nash equilibrium. This is a well written paper. It motivates the problem well and explains clearly why the proofs provided in the paper are important. No results are presented about the practical applications of the solution. It will be useful to explain the applications better and show some results. I appreciate the context provided in the outline section, though. Perhaps some examples of a few practical applications will benefit the paper. It appears that computing the denominator in equation 2.6 is computationally expensive for large problem. A comment about it would be helpful.